# Structure of the MRAS–SHOC2–PP1C phosphatase complex

Zachary J. Hauseman[1,5 ✉], Michelle Fodor[1,5], Anxhela Dhembi[1], Jessica Viscomi[1], David Egli[2], Melusine Bleu[2], Stephanie Katz[2], Eunyoung Park[3,4], Dong Man Jang[3,4], Kathryn A. Porter[1], Fabian Meili[1], Hongqiu Guo[1], Grainne Kerr[2], Sandra Mollé[2], Camilo Velez-Vega[1], Kim S. Beyer[2], Giorgio G. Galli[2], Saveur-Michel Maira[2], Travis Stams[1], Kirk Clark[1], Michael J. Eck[3,4], Luca Tordella[2 ✉], Claudio R. Thoma[1 ✉] & Daniel A. King[1 ✉]

RAS–MAPK signalling is fundamental for cell proliferation and is altered in most human cancers[1–3]. However, our mechanistic understanding of how RAS signals through RAF is still incomplete. Although studies revealed snapshots for autoinhibited and active RAF–MEK1–14-3-3 complexes[4], the intermediate steps that lead to RAF activation remain unclear. The MRAS–SHOC2–PP1C holophosphatase dephosphorylates RAF at serine 259, resulting in the partial displacement of 14-3-3 and RAF–RAS association[3,5,6]. MRAS, SHOC2 and PP1C are mutated in rasopathies—developmental syndromes caused by aberrant MAPK pathway activation[6–14]—and SHOC2 itself has emerged as potential target in receptor tyrosine kinase (RTK)–RAS-driven tumours[15–18]. Despite its importance, structural understanding of the SHOC2 holophosphatase is lacking. Here we determine, using X-ray crystallography, the structure of the MRAS–SHOC2–PP1C complex. SHOC2 bridges PP1C and MRAS through its concave surface and enables reciprocal interactions between all three subunits. Biophysical characterization indicates a cooperative assembly driven by the MRAS GTP-bound active state, an observation that is extendible to other RAS isoforms. Our findings support the concept of a RAS-driven and multi-molecular model for RAF activation in which individual RAS–GTP molecules recruit RAF–14-3-3 and SHOC2–PP1C to produce downstream pathway activation. Importantly, we find that rasopathy and cancer mutations reside at protein–protein interfaces within the holophosphatase, resulting in enhanced affinities and function. Collectively, our findings shed light on a fundamental mechanism of RAS biology and on mechanisms of clinically observed enhanced RAS–MAPK signalling, therefore providing the structural basis for therapeutic interventions.

The RAS family of proteins consists of highly homologous membrane-associated small GTPases that, after stimulation by RTK, or in disease contexts through activating mutations, switch to an active GTP-bound state resulting in conformational changes that enable interactions with various effector proteins[1,2]. RAF proteins are arguably the most important RAS effectors that, once activated, initiate the mitogen-activated protein kinase (MAPK) signalling cascade to drive cell proliferation and survival[3]. Given the fundamental cell processes controlled, the RAS–MAPK pathway is frequently hijacked through activating mutations in cancer cells[1–3], but also in clinically related developmental syndromes termed rasopathies[19]. Under normal conditions, signalling is tightly controlled at various levels to keep RAF kinases in an autoinhibited conformation through interactions with a 14-3-3 dimer and two regulatory RAF phospho-residues. Dephosphorylation of Ser259 of CRAF (Ser365 of BRAF and Ser214 of ARAF) by the MRAS–SHOC2–PP1C

holophosphatase is believed to result in the partial dissociation of 14-3-3, enabling RAF–RAS binding, and represents a crucial step in MAPK signal initiation[3–6]. Although we can appreciate the structural compositions and conformations of RAF-inhibited and RAF-activated complexes[4], structural resolution for the MRAS–SHOC2–PP1C complex involved in RAF activation is currently lacking. SHOC2 is a leucine-rich-repeat (LRR) domain–containing scaffold protein of uncharacterized structure that is thought to localize protein phosphatase 1 catalytic subunit (PP1C) to the membrane and possibly provide substrate selectivity to this otherwise promiscuous enzyme. SHOC2–PP1C is recruited to the cell membrane by MRAS, although several reports have shown binding to other RAS isoforms. Conflicting theories also exist on whether SHOC2 binds to RAS in a GDP- or GTP-bound state[5,6,12,20,21]. The physiological importance of this complex is highlighted by activating mutations in MRAS[7], SHOC2[8,10,12,13], PP1C[9,14] as well as CRAF (clustered around Ser259)[11] found

[1]Novartis Institutes for BioMedical Research, Cambridge, MA, USA. [2]Novartis Institutes for BioMedical Research, Basel, Switzerland. [3]Department of Biological Chemistry and Molecular Pharmacology, Harvard Medical School, Boston, MA, USA. [4]Department of Cancer Biology, Dana-Farber Cancer Institute, Boston, MA, USA. [5]These authors contributed equally: Zachary J. Hauseman, Michelle Fodor. ✉e-mail: zachary.hauseman@novartis.com; luca.tordella@novartis.com; claudio.thoma@novartis.com; dan.king@novartis.com

in Noonan-like syndrome, a type of rasopathy. Furthermore, SHOC2 depletion—on its own or in combination with MEK inhibitors—has been recently proposed as a therapeutic approach for certain RAS-driven cancers[15–18]. To gain structural understanding of the MRAS–SHOC2–PP1C holophosphatase responsible for RAF-activation and interpret clinically relevant mutations affecting its components, we aimed to characterize the subunit association and determine the structure of the complex. Our results reveal a highly cooperative assembly that is dependent on the nucleotide-loading state of MRAS, consistent with SHOC2 acting as a RAS effector. Rasopathy mutations of SHOC2, MRAS and PP1C map to subunit interfaces and enhance complex formation, driving RAF dimerization and MAPK flux. Of notable mechanistic importance, GTP-dependent recruitment of SHOC2 can be extended to canonical RAS isoforms, which form a ternary complex with reduced cooperativity compared with MRAS but with similar activity in vitro. Interestingly, we found significant co-dependencies of SHOC2 and canonical RAS proteins in cancer cells expressing oncogenic RAS with GTP-activating mutations but no co-dependency to MRAS in this setting. Taken together, these insights suggest that MRAS acts as the preferred component of the SHOC2–RAS–PP1C holophosphatase, but other RAS isoforms can substitute (especially when bearing a GTP-locking mutation).

## MRAS–SHOC2–PP1C cooperatively assembles

As a prelude to the structural analysis, we studied the assembly of the ternary SHOC2 complex from its constituent components. We individually expressed and purified recombinant SHOC2 (residues 80–582, corresponding to the conserved LRR domain), MRAS (residues 1–178) and PP1Cα (residues 7–300) and studied their interactions using surface plasmon resonance (SPR). To evaluate its pairwise interactions, we immobilized MRAS (loaded with either GDP or the non-hydrolysable GTP analogue GppNHp) and performed SPR analysis using SHOC2 or PP1Cα as the analyte. Whereas SHOC2 bound to MRAS in a concentration- and nucleotide-dependent manner, PP1Cα did not, even at high micromolar concentrations (SHOC2, dissociation constant ($K_D$) = 6.12 μM; PP1Cα, $K_D$, no binding; Fig. 1a and Extended Data Fig. 1a,b). We were also unable to detect binding of PP1Cα to immobilized SHOC2 (Extended Data Fig. 1c), indicating that PP1Cα does not form individual binary complexes with MRAS or SHOC2 under these conditions. Notably, the affinity of SHOC2 for immobilized MRAS was increased 70-fold in the presence of an excess of PP1Cα. These findings suggest that PP1Cα requires the simultaneous presence of MRAS and SHOC2 to participate in the ternary complex, whereas SHOC2 and MRAS can form a weaker, independent binary complex. The strong positive cooperativity for SHOC2–MRAS binding in the presence of PP1Cα indicates that the ternary complex involves reciprocal interactions among the component proteins, rather than independent scaffolding of MRAS and PP1C by SHOC2.

We reasoned that rasopathy-associated gain of function (GOF) mutations in SHOC2 and/or MRAS might stabilize the ternary complex and therefore facilitate structural analysis. SHOC2 with an M173I substitution, which is found in some individuals presenting with features of Noonan syndrome[8], displayed a modest increase in effective affinity in our SPR assay (2.5×/5× binary/ternary; Fig. 1a). Similarly, the rasopathy-associated Q71R mutation in MRAS, which is expected to inhibit nucleotide hydrolysis and stabilize the active GTP-bound conformation, also resulted in an increase in affinity to the wild-type (WT) SHOC2 LRR domain (8×/3× binary/ternary; Fig. 1a). The PP1Cα P50R mutation, which is a substitute for the PP1Cβ GOF P49R mutation, similarly improves the affinity of the ternary complex. Combining MRAS(Q71R) with SHOC2(M173I) further stabilized both the binary and ternary complexes (12×/7× binary/ternary; Fig. 1a). To test whether the unstructured SHOC2 N terminus could contribute to complex assembly, we purified full-length SHOC2 (residues 2–582). Whereas binary interactions with MRAS were equivalent or slightly inferior to SHOC2(80–582),

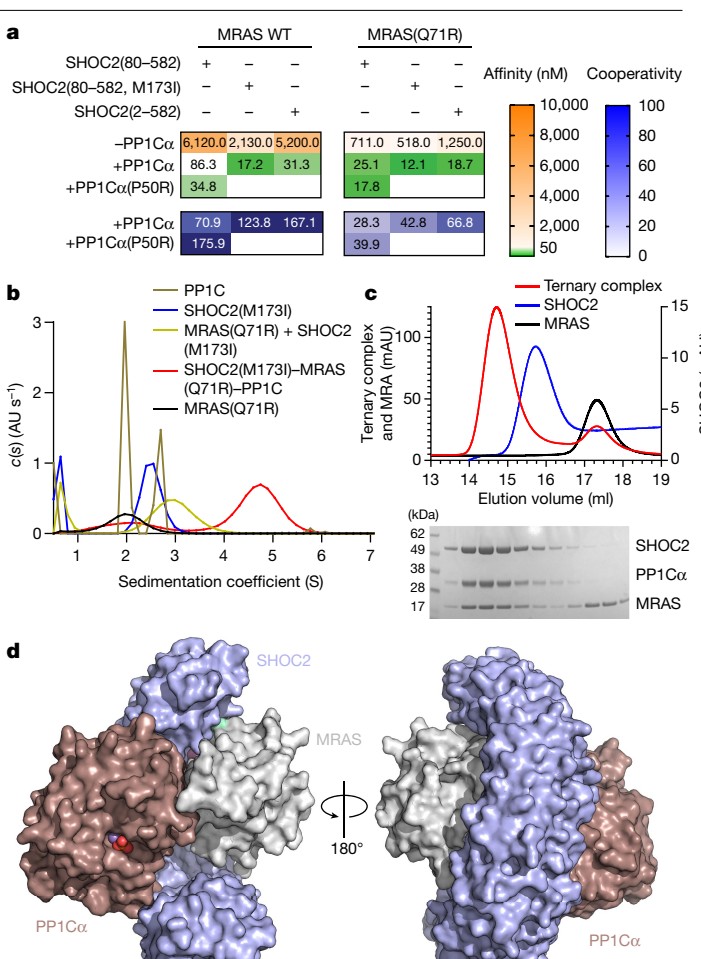

**Fig. 1 | Cooperative assembly of the SHOC2–MRAS–PP1Cα ternary complex. a**, The affinity ($K_D$) and associated cooperativity ($\alpha$) values derived from SPR sensorgrams with surface-immobilized MRAS–GppNHp (WT or Q71R) and the indicated analytes. Cooperativity is defined as the ratio of binary and ternary $K_D$ values, $\alpha = K_D^{SHOC2}/K_D^{SHOC2-PP1C}$. The increase in response units (RU) signal is consistent with the saturation of MRAS with both proteins $R_{max}$, SHOC2, ~800–1,000 RU; $R_{max}$, SHOC2/PP1C, ~1,250–1,500 RU). **b**, Sedimentation coefficient distributions $c(s)$ derived from sedimentation velocity profiles of MRAS(Q71R) (black); PP1Cα (bronze); SHOC2(80–582, M173I) (blue); mixture of MRAS(Q71R) and SHOC2(80–582, M173) (gold); mixture of SHOC2(80–582, M173I), MRAS(Q71R) and PP1Cα (red). The raw sedimentation signal was acquired over time by absorbance at 280 nm at a rotor speed of 42,000 rpm at 20 °C. All proteins were sedimented at equimolar concentrations of 10 μM. Data are representative of two independent experiments. **c**, Ultraviolet light absorbance at 280 nm traces from semi-preparative SEC (top). SDS–PAGE analysis of individual fractions corresponding to the ternary complex trace (bottom). The SEC experiment was independently repeated three times with similar results. **d**, Surface representation of the ternary complex of SHOC2 (pale blue), MRAS (grey) and PP1Cα (maroon). The phosphate group bound at the active site is shown as red spheres, and Mn²⁺ is shown as purple.

ternary complex formation was improved, consistent with the possibility of a direct PP1Cα interaction with the N terminus of SHOC2. Both analytical ultracentrifugation and semi-preparative size-exclusion chromatography (SEC) confirmed that equimolar ratios of MRAS(Q71R), SHOC2(80–582, M173I) and PP1Cα could efficiently form a stable complex (Fig. 1b,c). A complex comprising the M173I rasopathy mutant of the SHOC2 LRR domain (80–582), GTPase-deficient MRAS(Q71R) and WT PP1Cα was isolated and crystalized, diffracting to 1.95 Å ($R_{cryst}$ = 0.1838; $R_{free}$ = 0.2209; Fig. 1d, Methods and Extended Data Table 1).

## MRAS–SHOC2–PP1C holophosphatase structure

The holoenzyme exhibits a compact structure in which C-shaped SHOC2 wraps around both MRAS and PP1Cα, enabling reciprocal interactions between all three subunits (Fig. 2a). SHOC2 exhibits the expected LRR fold; the 20 repeats of SHOC2 coil to form a solenoid with an extended β-sheet along its inner, concave surface. Both GppNHp-bound MRAS and PP1Cα are present as globular domains in their active states, in agreement with published apo structures[22–24]. Highly conserved residues on the concave surface of SHOC2 engage MRAS and PP1Cα (Extended Data Fig. 1d,e). MRAS is engaged primarily by the descending loop and β-strands of LRR domains 2–10, whereas PP1Cα is coordinated by the ascending loops just C-terminal to the β-strand of each LRR spanning nearly the entire SHOC2 solenoid (Fig. 2b,c). We also determined the structure of the isolated LRR domain of SHOC2 (residues 80–582) at 1.9 Å resolution (Extended Data Fig. 2a). The unliganded domain adopts a similar overall structure, but with a modest change in its overall curvature compared with SHOC2 in the ternary complex (Extended Data Fig. 2b). This degree of flexibility is seen for many LRR proteins. In both the apo structure and ternary complex, the SHOC2 β-sheet is disrupted between strands 9 and 10, with bridging water molecules inserted into the otherwise regular hydrogen-bonding pattern between the parallel β-strands (Extended Data Fig. 2c). This appears to be a conserved feature of SHOC2, as it is accurately predicted by Alphafold[25].

The MRAS GTPase adopts an active conformation, and clear density is observed in the structure for the bound GTP analogue. MRAS switch I (SWI) and switch II (SWII) loops that rearrange in the GTP- versus GDP-bound states are at the core of the complex. In the present structure, both SWI and SWII of MRAS interact extensively with SHOC2 (Fig. 2b–d), while SWI, together with MRAS helix α1 and the adjacent regions, simultaneously engage PP1Cα. SWI and SWII mediate most of the interactions recruiting the N-terminal LRR domains of SHOC2 to MRAS. Both electrostatic and hydrophobic interactions contribute to the interface, which also contains 16 coordinated waters. Notably, bound MRAS(Q71R) sits in an especially 'closed', active GppNHp conformation that differs substantially from the unbound form of GppNHp–MRAS and NRAS[22,26] (Protein Data Bank (PDB): 1X1S, 1X1R, 5UHV; Extended Data Fig. 3a,b). Importantly, the rearrangement of MRAS SWII enables additional interactions with SHOC2. This conformational shift enables SWII residues to mediate direct interactions and foster a coordinated solvent network at the interface. Direct interactions, highlighted by MRAS residues Gln80 and Glu73 and SHOC2 residues Asp106 and Asn265, are supported by numerous water-mediated interactions between the main chain of SWII and the β-sheet of SHOC2 (Fig. 2e).

The SWI motif of unbound GppNHp–MRAS also adopts an open, inactive conformation but we found that the bound MRAS SWI mirrors that of the active canonical RAS isoforms (Extended Data Fig. 3a,b). This active conformation is required for proper orientation of SWI residues at the SHOC2 interface while simultaneously supporting PP1Cα binding through SWI residues Glu47 and Asp48. These MRAS SWI and adjacent residues interact with PP1Cα residues surrounding helix G of the small helical domain on the opposite surface of the active site. The interaction between SWI residues and the surrounding solvent interaction network extending directly to the SHOC2 interface (Fig. 2d and Extended Data Fig. 3c) highlights the MRAS conformational shift (relative to apo MRAS) that is critical to both SHOC2 and PP1Cα binding.

PP1Cα binds to an extensive composite surface formed by SHOC2 and MRAS (Extended Data Fig. 4a,b) and is oriented such that its active site is exposed and fully accessible for substrate binding in the holoenzyme complex (Fig. 3a). PP1C functions as the catalytic subunit for diverse PP1 phosphatase complexes, and it is highly conserved across phylogeny. PP1C residues that contact SHOC2 and MRAS are almost identically conserved in the three isoforms (α, β and γ) expressed in higher eukaryotes (Extended Data Fig. 4c). Thus, we expect that any of the three isoforms

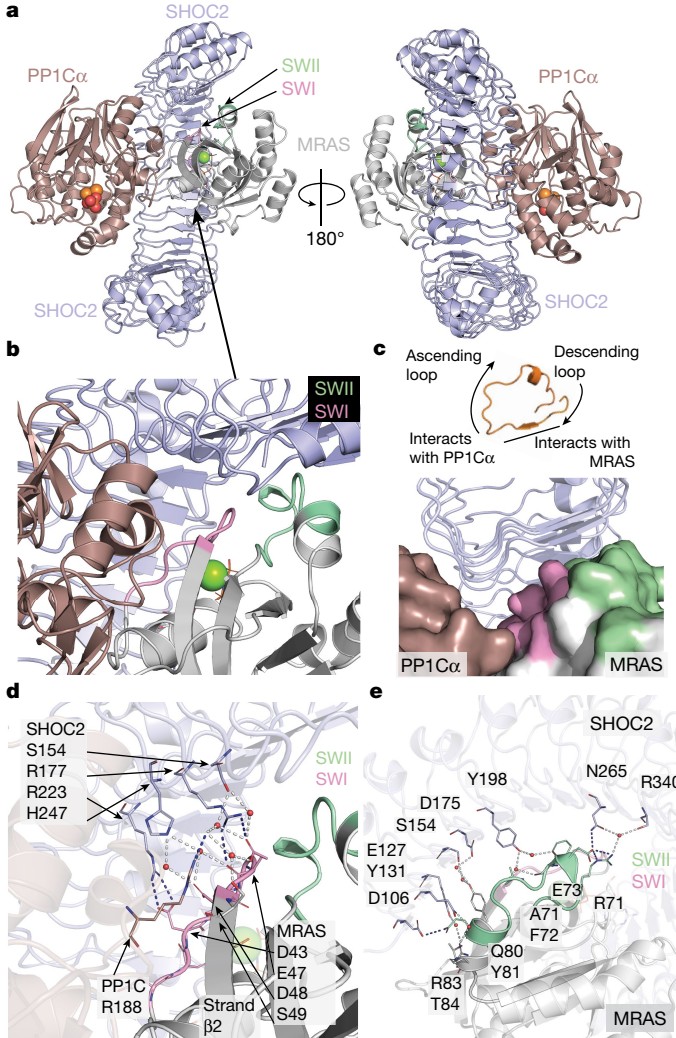

**Fig. 2 | SHOC2–MRAS–PP1Cα interfaces include polar contacts and coordinated solvent. a**, Cartoon view of the complex of SHOC2 (pale blue), MRAS (grey) and PP1Cα (maroon), illustrating interactions mediated by the concave surface of SHOC2. SWI and SWII of MRAS are highlighted in pink and green, respectively; GppNHp is rendered in ball and stick; and Mg²⁺ is represented as a green sphere. The phosphate bound to the active site of PP1C is represented in red, and the Mn²⁺ metals are indicated as orange spheres. This colour scheme is maintained throughout all of the figures. **b**, Cooperative interface of SHOC2, PP1Cα and MRAS mediated by SWI (pale green) and SWII (pink) of MRAS. **c**, SHOC2 uses its β-sheet surface and descending loops from multiple LRRs to recruit MRAS while binding to PP1Cα through the ascending loops of numerous LRRs. One representative LRR (orange) is shown. MRAS and PP1Cα are shown in surface representation. **d**, Direct and water-mediated interactions at the cooperative interface highlighting MRAS SWI interactions with Arg188 of PP1Cα helix G. The indicated side chains are colour coded by subunit with representative water molecules indicated by red spheres and bonding networks highlighted by dashed lines. An MRAS-bound magnesium ion is shown as a green sphere. **e**, Direct and water-mediated interactions at the cooperative interface as in **d**, but highlighting MRAS SWI and SWII interactions with SHOC2.

can assemble to form the SHOC2 holoenzyme. PP1 activity is governed through its interaction with many regulatory proteins that target the phosphatase to specific cellular locations and determine substrate specificity[23,24,27–33]. Typically, these regulators bind to PP1Cα through short linear motifs of 4–8 amino acids found within intrinsically disordered regions of the protein. Common binding sites for a number of these regulatory partners have been mapped onto the surface of PP1C, distal from the acidic, hydrophobic and C-terminal substrate-binding

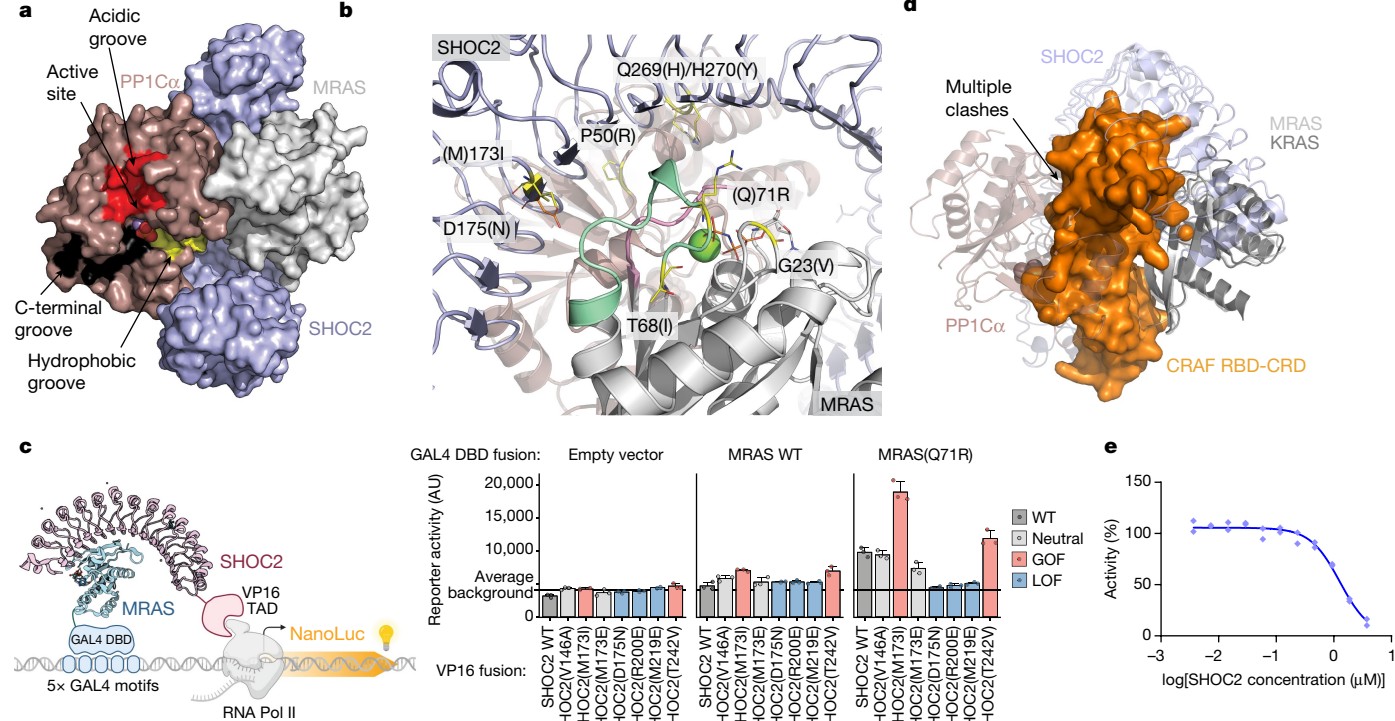

**Fig. 3 | Rasopathy mutations and RAF binding predicted to affect complex formation. a**, Surface representation of the SHOC2–PP1Cα–MRAS holoenzyme highlighting the active site and accessible acidic (red), C-terminal (black) and hydrophobic (yellow) substrate-binding grooves of PP1Cα. The RVXF and other regulatory surfaces of PP1Cα are not occupied in the SHOC2(80–582)–MRAS(Q71R, 1–178)–PP1Ca(7–300) ternary complex. **b**, GOF rasopathy mutations localize to PPI interfaces. The indicated GOF and LOF rasopathy mutations are shown as sticks highlighted in yellow and orange, respectively; SHOC2 and MRAS are shown in grey with SWI and SWII indicated. SHOC2 Ile173 (Met173 in WT SHOC2) fills a hydrophobic space produced by SWI, SWII and SHOC2 LRR domains. SHOC2 Gln269/His270 resides at the interface of all three subunits, whereas SHOC2 Asp175 coordinates a water to MRAS Tyr81. PP1Cα Pro50 sits at the PPI interface but cannot make substantial interactions owing to the limitations of the side chain. **c**, Schematic of the M2H

system (left). The Nluc reporter signal resulting from concurrent expression and interaction between GAL4(DBD)–MRAS and SHOC2–VP16(TAD) of WT and mutant variants (right). Data are mean ± s.e.m. *n* = 3 independent biological replicates. The diagram was created with BioRender.com. **d**, Alignment of the MRAS–SHOC2–PP1C ternary holophosphatase structure with a structure of KRAS bound to the CRAF RBD-CRD domain (PDB: 6XHA). Note that the alignment (using RAS as the key object) results in multiple clashes between RBD-CRD and PP1C–SHOC2. **e**, Untagged full-length SHOC2 disrupts the interaction of MRAS(Q71R)–GppNHp and RBD as assessed in a TR-FRET assay (IC$_{50}$ 913nM), indicating that SHOC2 and RBD binding to RAS is incompatible. Technical replicates are shown as individual data points; one of three independent experiments is shown. Competition IC$_{50}$ values for additional tested combinations are provided in Extended Data Table 3.

grooves. All of these substrate-binding grooves are accessible on PP1C in the SHOC2 holoenzyme (Fig. 3a). The binding mode of SHOC2 is more reminiscent to that of SDS22[31,32], a regulatory partner that is conserved from yeast to humans and binds to PP1C through extensive interactions of its LRR domain. Although their binding sites partially overlap, there is no direct structural correspondence in the binding of SHOC2 and SDS22 to PP1C. The interactions between SHOC2 and PP1Cα are predominantly polar, and the binding interfaces among all three subunits contain a notable amount of ordered solvent. Nonetheless, the extensive inter-subunit interactions bury a total of around 4,966 Å² of solvent-accessible surface. Of this total, only 1,762 Å² are buried between SHOC2 and PP1Cα (around 881 Å² on the surface of each). The highly polar and solvated nature of the SHOC2–PP1Cα interface underlies the very low affinity of their interaction in the absence of MRAS.

Overall, our high-resolution X-ray structure of the MRAS–SHOC2–PP1Cα holophosphatase sheds light on cooperative complex assembly and enables a detailed dissection of the interacting amino acids between the three components.

## Rasopathy mutations localize to PPIs

GOF mutations causing rasopathies have been reported on SHOC2, MRAS and PP1Cβ[7–10,13,14,19,34,35]. Moreover, several SHOC2 loss of function

(LOF) mutants unrelated to rasopathy were identified through genetic screens that abolish RTK–RAS-activated phenotypes in *Caenorhabditis elegans*[20,36,37]. We mapped the locations of these various mutants onto our structure. Notably, almost all reported GOF mutations localize to PPI interfaces in the SHOC2–MRAS–PP1Cα structure, with the majority co-localizing to the SWI/SWII-mediated cooperative interface with SHOC2 and PP1Cα (Fig. 3b). Rasopathy-associated mutations SHOC2(M173I) and Q269_H270delinsHY (Q269H/H270Y); MRAS G23V, T68I and Q71R; PP1Cα P50R (corresponding to PP1Cβ P49R); and non-rasopathy SHOC2 LOF mutations D175N and E457K all map to interaction sites. These observations support the conclusion that GOF rasopathy mutations could increase MAPK signalling by modulating the surfaces that drive complex association.

SHOC2(M173I), which shows a mild increase in affinity towards WT MRAS in our SPR as well as in cellular two-hybrid assays (discussed below), is resolved in our structure and lies at the surface of a critical hydrophobic patch formed by SHOC2 Thr150, Asp175 and Thr196, and SWII MRAS residues Ala76 and Met77. Moreover, the affinity of the SHOC2(M173I)–MRAS(Q71R)–PP1Cα(WT) complex is clearly higher than that of the SHOC2(WT)–MRAS(WT)–PP1Cα(WT) complex as measured by SPR (Fig. 1a). To rationalize the latter difference in affinity from a structural standpoint, we ran molecular dynamics simulations of our mutant structure and the corresponding WT ternary complex. For the

WT complex, we back-mutated SHOC2 Ile173 and MRAS Arg71 on our structure to the corresponding WT residues (SHOC2 Met173 and MRAS Gln71). Although the conformational ensemble obtained from molecular dynamics simulations of our mutant structure closely resembles the crystallographic state, the simulations of the WT complex show three distinct states of the MRAS–SHOC2 interface (Extended Data Fig. 5a). The dominant state of the WT simulations (67% of the ensemble) shows direct protein–protein hydrogen bonds between MRAS and SHOC2 in the vicinity of Met173, whereas the less prevalent states (33% of the ensemble) show water-mediated hydrogen bonding around this region. This local loss of 'tightness' between MRAS WT and SHOC2 WT may entail an entropic cost resulting from an increased presence of water at the interface, providing support for the lower affinity observed in the SPR analysis for the SHOC2(WT)–MRAS(WT)–PP1Cα(WT) complex relative to the SHOC2(M173I)–MRAS(Q71R)–PP1Cα(WT) complex. We next examined the effect that the GOF and LOF mutations had on the SHOC2–MRAS interaction in cells. We developed a mammalian two-hybrid (M2H) assay in which the GAL4 DNA-binding domain was fused to MRAS WT or MRAS(Q71R), and we monitored luciferase activity proportional to SHOC2–VP16/TAD binding (Fig. 3c). Given that a substantial fraction of WT MRAS in cells was in the GDP state, the MRAS(Q71R) mutation was used to lock GTP-induced RAS conformational changes, which we found was necessary to detect strong interactions with SHOC2. We investigated the validity of our M2H assay by mutating critical SHOC2–MRAS interacting residues inferred from our X-ray ternary structure. We found that mutation of SHOC2 away from the PPIs (V146A) does not alter the SHOC2–RAS interaction. By contrast, SHOC2 mutations at the MRAS SWI–SWII interface (R200E or M219E) resulted in the loss of SHOC2–MRAS binding, and T242V produced a notable gain of affinity. We next showed that rasopathy-derived SHOC2 GOF mutant M173I results in a twofold increase in MRAS recruitment, which is reverted when the isoleucine is replaced by a glutamic acid. In the same hot-spot PPI region, LOF mutation D175N abolished the binding between SHOC2 and MRAS (Fig. 3c). Given the non-native (nuclear) localization of the proteins in this assay signal, we used an orthogonal SHOC2–eGFP bipartite system which enabled quantitative assessment of SHOC2–eGFP abundance compared with an mCherry expression control. We show that mutations did not alter SHOC2 abundance or result in aggregation in cells (the precise distribution in cellular compartments was not analysed) (Extended Data Fig. 6a,b).

Overall, our findings here validate the structural observations and reveal that disease-relevant mutations, such as those found in rasopathies, localize to PPIs and modulate complex assembly to affect the function of the resulting complex.

## SHOC2 and RAF RBD compete for RAS binding

MRAS SWI residues at the core of the SHOC2–PP1Cα interface are also responsible for recruitment of the RAS-binding and cysteine-rich domains (RBD-CRD) of RAF to the plasma membrane. We note that C-terminal residues of SWI extending into strand β2 are critical to both the RBD and PP1Cα interfaces. Alignment of MRAS in the phosphatase ternary complex to KRAS of the previously solved KRAS–RAF[RBD-CRD] co-structure is incompatible with the recruitment of RAF by SWI of this MRAS subunit into a single holoenzyme complex (PDB: 6XHA; Fig. 3d), suggesting that RAS cannot simultaneously engage SHOC2–PP1Cα and RAF kinases. To confirm this experimentally, we first measured the affinity of MRAS to the isolated cRAF1 RBD (hereafter RBD) by SPR and confirmed that the affinities are similar between the MRAS WT and Q71R mutant and comparable to the canonical NRAS(Q61R) variant (Extended Data Table 2).

To confirm that RBD binding would be incompatible with SHOC2-mediated complex formation, we established a time-resolved fluorescence energy transfer (TR-FRET) assay between RBD and GppNHp-loaded MRAS(Q71R) and assessed the ability of untagged

full-length SHOC2 (residues 2–582) to disrupt this interaction. Consistent with the expectation of competition with RBD binding for RAS, we obtained full inhibition in this assay with a half-maximum inhibitory concentration ($IC_{50}$) of 913 nM (Fig. 3e), consistent with the affinity of SHOC2 to MRAS(Q71R). To confirm and further expand on this initial result, we wanted to assess whether this observation is transferrable to other RAS proteins and established the analogous competition assays with MRAS WT and NRAS(Q61R) and included the GOF mutant SHOC2(M173I) (Extended Data Table 3). As in the MRAS(Q71R)–RBD setting, full-length SHOC2 also displaced RBD from MRAS WT with $IC_{50}$ values in line with the SPR data. The higher affinity of SHOC2(M173I) to MRAS (as observed in SPR) was also recapitulated in the TR-FRET system, showing a lower $IC_{50}$ for SHOC2(M173I) compared with SHOC2 WT. Interestingly, SHOC2 also showed very similar competition towards NRAS(Q61R)–RBD suggesting that SHOC2 also interacts with canonical RAS proteins (Extended Data Table 3).

## NRAS and KRAS form functional holophosphatases

MRAS residues that interact with SHOC2 are highly conserved across phylogeny in non-canonical MRAS and RRAS as well as the canonical RAS proteins (Extended Data Fig. 7a). By contrast, a subset of MRAS residues that contact PP1C are not conserved in RRAS1, RRAS2 or the canonical RAS proteins. These unique interactions may underpin the preferential assembly of active MRAS in the holophosphatase complex. To mechanistically investigate whether the canonical RAS isoforms can bind to SHOC2 and form a ternary complex, we performed SPR analysis using NRAS and KRAS bound to SPR chip surfaces (Fig. 4a). We found that GppNHp-loaded NRAS and KRAS are capable of binary interaction with SHOC2 at levels comparable to MRAS (Fig. 1a), consistent with the high degree of conservation of sequences responsible for binding to SHOC2. Notably, although we could observe the clear formation of the ternary complex after addition of PP1C, the cooperativity of the interaction was almost completely lost. We found that these NRAS- and KRAS-containing ternary complexes exhibit between 4-fold and 33-fold lower overall affinity compared with the MRAS counterparts. We further observed that GTP-locking Q61R mutations positively affect NRAS- and KRAS-containing complexes compared with GppNHp-loaded WT proteins. This suggests that, like in MRAS, the Q61R mutation could add a residue that can not only rigidify active conformations but also actively participate in the RAS–SHOC2 PPI. Although these mutations significantly improved overall complex affinity, this could not fully restore binding to the levels of the complex containing WT MRAS. To directly test whether MRAS can outcompete other RAS isoforms for holophosphatase binding, we mixed recombinant KRAS and MRAS together with one equivalent each of SHOC2 and PP1Cα, and purified the complex using SEC. We found that the KRAS-containing complex can be readily isolated (Extended Data Fig. 7b), but MRAS is able to fully outcompete KRAS under equimolar conditions. This indicates that GTP-bound MRAS is the optimized RAS isoform within the holophosphatase, whereas the other RAS isoforms can substitute in the absence of MRAS.

To test whether our purified complex was active and specific for the RAF Ser259 phosphorylation site, we purified complexes containing MRAS, PP1Cα and WT, or the GOF SHOC2(M173I) mutant. We found that all holophosphatase complexes were capable of site-specific dephosphorylation of recombinant BRAF in an active, dimeric state. The assembled complex specifically dephosphorylated BRAF Ser365 (the equivalent to CRAF Ser259) while showing no activity towards BRAF's C-terminal 14-3-3-binding site Ser729 (the equivalent to CRAF Ser621) in vitro. The rasopathy GOF SHOC2(M173I) mutation boosted this activity (Fig. 4b). Furthermore, we could show that complex containing PP1Cα, SHOC2(M173I) and KRAS instead of MRAS was still capable of this selective dephosphorylation, indicating that, once formed, the holophosphatase activity is not dependent on RAS isoform.

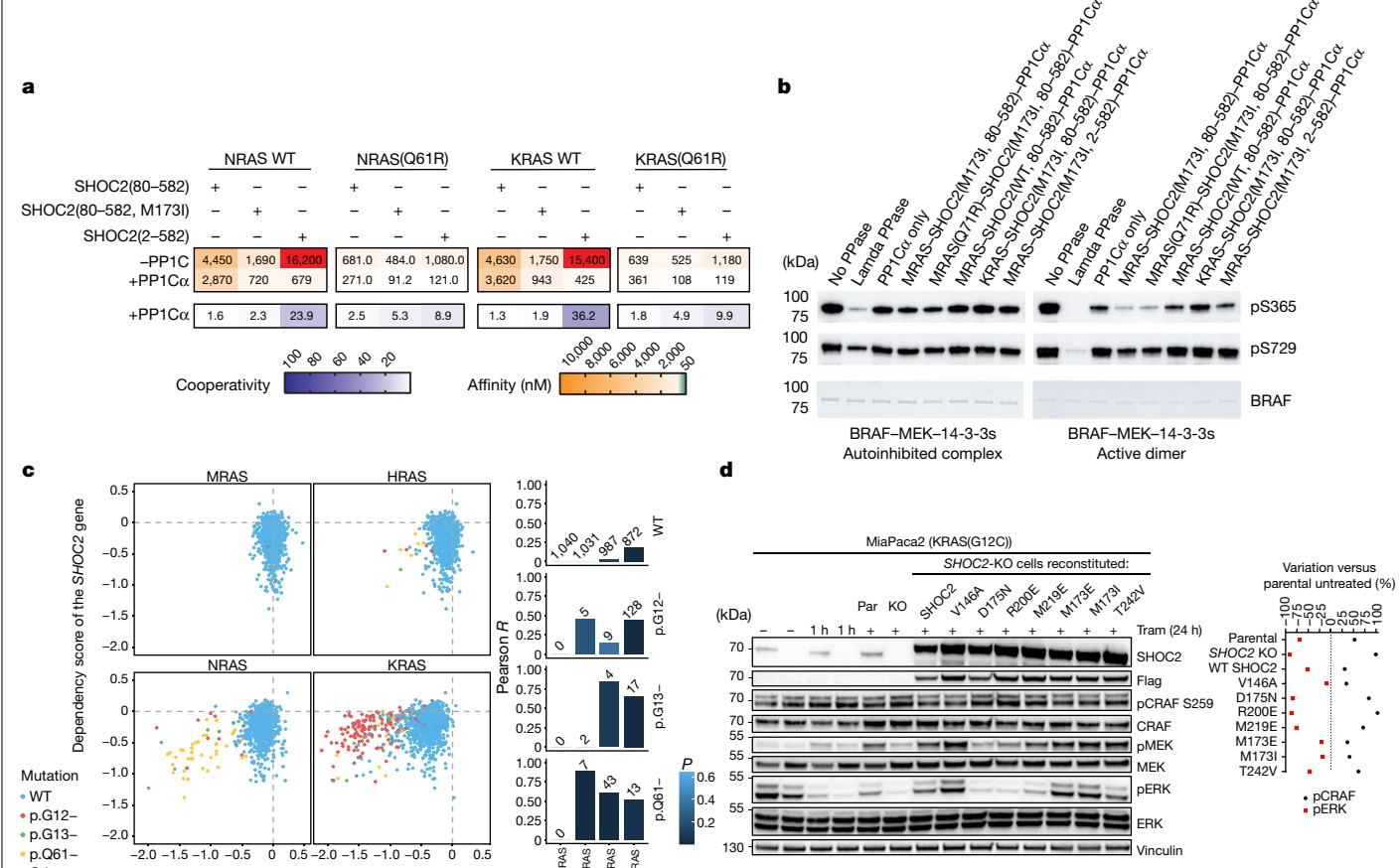

**Fig. 4 | Canonical RAS isoforms can form active holophosphatases. a**, SPR-derived affinity and cooperativity immobilizing NRAS–GppNHp, NRAS(Q61R)–GTP, KRAS–GppNHp or KRAS(Q61R)–GTP with the indicated analytes. $\alpha = K_D^{SHOC2}/K_D^{SHOC2-PP1C}$. Affinity values over 10,000 nM are highlighted in red. **b**, In vitro dephosphorylation of autoinhibited and active-state BRAF–MEK1–14-3-3 complexes by the SHOC2 holophosphatase. Purified full-length BRAF complexes in the autoinhibited state (left) or active dimeric state (right) were incubated with lambda phosphatase (PPase), PP1Cα or ternary SHOC2 complexes, and were blotted with phospho-specific antibodies for BRAF phosphorylated at Ser365 and Ser729. Equivalent loading of BRAF complexes is shown by Coomassie staining. Phosphorylated Ser365 is selectively dephosphorylated relative to Ser729 in the active dimer, whereas both are relatively protected in the autoinhibited (14-3-3-bound) state. Experiments were conducted twice with similar results. **c**, The relationship between the knockdown of *SHOC2* and *MRAS*, *HRAS*, *NRAS* and *KRAS*. The dependency scores of each *RAS* gene and *SHOC2* are shown on the *x* and *y* axes, respectively. Dashed lines indicate a dependency score of zero. A highly negative dependency score implies that a given cell line is highly dependent on that gene. Cell lines dependent on both SHOC2 and RAS are indicated at the bottom left. Right, the calculated Pearson correlation coefficient (*y* axis) applied to each mutation group (group of cell lines containing the associated mutation). A higher positive value indicates a stronger positive relationship: the dependency score of SHOC2 decreases/increases in the same lines as the dependency scores of the RAS genes. The *n* values above each bar show the number of cell lines in each mutation group. **d**, Immunoblot analysis of MiaPaca2 parental cells (Par), *SHOC2*-KO (KO) and stable cell lines reconstituted with SHOC2 mutants after 10 nM trametinib (Tram) treatment for 1 h or 24 h (+). Densitometry quantification (percentage variation) of pCRAF/CRAF and pERK/ERK levels from the immunoblot analysis normalized to untreated MiaPaca2 parental cells. The samples were derived from the same experiment and blots were processed in parallel. The images are representative of two independent experiments.

Together our data show that RAS mutations that favour GTP-locked states can facilitate complex formation and, independently, that RAS canonical isoforms can substitute for MRAS. Notably, in cancer, canonical RAS proteins but not MRAS are frequently subjected to activating mutations that drive cancer cell survival and proliferation. An analysis of a large dataset of cancer functional screens (more than 1,000 cell lines) available from DepMap (https://depmap.org/portal/) shows that SHOC2 is necessary for the survival of cell models carrying GTP-activating mutations in canonical RAS proteins, underlying functional interaction. Interestingly MRAS depletion has no influence on cell fitness in models in which SHOC2 is essential (Fig. 4c). SHOC2 and RAF1 (CRAF) dependency profiles correlate well across all cell models, highlighting their fundamental functional interaction, which is independent of a given RAS partner (Extended Data Fig. 7c).

To validate these findings, we performed experiments in KRAS[G12C]-mutated MiaPaca2 cells in the presence of the MEK inhibitor trametinib.

SHOC2 has shown synthetic lethality in *KRAS*-mutant cancer lines, such as in MiaPaca2 cells treated with MEK inhibitors, which, after initial pathway inhibition, cause pathway rebound and increased RAS GTP loading[18]. Consistent with these reports, we found that knockout (KO) of endogenous *SHOC2* prevents MAPK pathway reactivation (24 h trametinib treatment) resulting in increased phosphorylated CRAF Ser259 (inhibited RAF) and reduced phosphorylated ERK and phosphorylated MEK, which measure MAPK-pathway downstream activity (Fig. 4d). Under the same conditions, knockdown of *MRAS* expression did not produce any significant change in the phosphorylation of CRAF Ser259 nor downstream markers (Extended Data Fig. 7d; compare lanes 1–2 for *MRAS* knockdown with lanes 3–4 for *SHOC2* KO). These and previous results agree with the data from cancer functional screens and suggest that, in RAS–GTP-locking conditions, such as those resulting from oncogenic mutations, canonical RAS proteins can substitute for MRAS within the SHOC2–PP1C complex.

Finally, we used the context of the MiaPaca2 *SHOC2*-KO model to study the functional effects of *SHOC2* mutants on MAPK activation markers (Fig. 4d). Although reconstitution with WT SHOC2 and neutral or GOF mutations restored pCRAF (Ser259), pMEK and pERK levels to that of the parental cell line, SHOC2 LOF mutations such as D175N were comparable to the KO, indicating that SHOC2 holophosphatase association influences the activity of the MAPK pathway. Together with the previously presented interaction data in M2H (Fig. 3c), rational mutagenesis at the MRAS–SHOC2 PPI enabled us to direct the affinity and activity of these components in cells, confirming the fidelity of these interaction surfaces in our structure.

## Discussion

Here we characterized the assembly and structure of the SHOC2–MRAS–PP1Cα holoenzyme, which is thought to be critical to regulating the phosphorylation state of RAF and the activity of the MAPK pathway. Using individually purified subunits, we demonstrate that the components assemble with high cooperativity, displaying a 70× increase in the affinity of the LRR domain of SHOC2 for GppNHp-loaded MRAS in the presence of PP1Cα. Furthermore, we were able to demonstrate a stable interaction between PP1Cα and SHOC2 or MRAS and PP1Cα only in the presence of the third member of the holoenzyme. Our structure explains the observed cooperativity, whereby the SHOC2 LRR acts as lynchpin recruiting both MRAS and PP1Cα through an extensive concave surface while creating an interdependent protein–protein and water-mediated interaction network simultaneously involving all three subunits. GppNHp-loaded MRAS is at the core of the complex and SHOC2 behaves as a RAS effector molecule, interacting with both SWI and SWII in their active conformations, enabling SWI to interact with PP1Cα. SWI's central role at the interface, mediating interactions with both SHOC2 and PP1Cα, and our alignments and competition experimental data are not consistent with direct recruitment of the RBD-CRD and the downstream Ser259 phosphorylation site of RAF through the RAS protomer found in our complex. Our recruitment studies with full-length SHOC2 containing an intact N terminus show an increased affinity toward MRAS only in the presence of the phosphatase (Fig. 1a), suggesting a direct interaction of SHOC2's N terminus with PP1Cα. PP1Cα is frequently found in complex with multiple regulators through short linear interaction motifs and, while this Article was under review, a preprint was published with the crystal structure of the holoenzyme containing full-length SHOC2, displaying a direct interaction between the RVXV motif of SHOC2 and the RVXF-binding pocket of PP1Cα[38]. An existing conflict in the SHOC2 field is whether SHOC2 interacts exclusively with MRAS or promiscuously with multiple RAS isoforms. Original cellular immunoprecipitation studies identified SHOC2 and PP1C as effectors unique to MRAS. However, recent studies have displayed a synthetic lethal interaction of SHOC2 depletion, but not MRAS, with MEK inhibitors in *KRAS*-mutant pancreatic or lung cancer cell lines[18]. Our recruitment studies indicate that all of the RAS isoforms studied recruit SHOC2 with a similar affinity and show the same GTP-loading dependence found in other RAS effector interactions. However, the divergent RAS–PP1C interface appears to favour MRAS as a preferred component of this complex—as evidenced by the comparatively low affinity and cooperativity of ternary complexes bearing NRAS or KRAS[18]. DepMap analysis and our cellular experiments are consistent with the dispensability of MRAS suggesting that, in certain cellular contexts, MRAS could be replaced as the dominant RAS species in the holophosphatase complex, such as in presence of oncogenic mutations that significantly shift the GTP-loading state of these RAS proteins[39]. The exact contribution of each RAS isoform in the holoenzyme and the underlying biological context remains an outstanding question. Notably, the extensive interface mediated by the concave surface of SHOC2 and SWI of MRAS is similar to other PP1 or PP2A holoenzymes containing scaffolding subunits with HEAT (Huntingtin, elongation

factor 3, protein phosphatase 2A, yeast kinase TOR1) or LRR repeats. The three substrate grooves of PP1Cα and active-site metal ions point away from the core of the complex, whereas the interface of PP1Cα and MRAS extends the phosphatase's hydrophobic groove leading to the active site. Extension of the hydrophobic binding groove has been postulated as a substrate-recruitment mechanism of the similar PP2A holoenzyme containing both a scaffolding and regulatory subunit. It is possible that RAS has a similar recruitment mechanism for RAF phosphorylated at Ser259. Regardless, we demonstrate that the assembled holophosphatase has increased activity and specificity towards RAF Ser259 in vitro (Fig. 4b), suggesting direct recognition before dephosphorylation.

Together, the RAS GTP-binding specificity along with the mutually exclusive RAS interaction to either SHOC2–PP1C or RAF allows speculation of the existence of a multi-molecular RAS signalosome model with distinct complexes, in which at least two individual RAS molecules, after activation, simultaneously recruit RAF–14-3-3 and the holophosphatase to enable RAF activation. Whether such a coordinated mechanism requires direct interaction between the two distinct complexes through, for example, the RAS units, or simple co-localization at the cytoplasmic membrane remains to be established. Finally, we provide insights for the relevance of such active RAS–SHOC2–PP1C holophosphatase complexes in rasopathy and cancer settings. Our structural mapping of disease-relevant rasopathy mutations in SHOC2, MRAS and PP1C, as well as structure–function studies on specific SHOC2 GOF and LOF variants, establish a mechanistic basis for their function. Namely, the modulation of protein–protein interaction affinities translates into enhanced pathway activation when strengthening PPIs and reduced pathway activation when weakening PPIs in the holophosphatase complex, respectively. Collectively these results provide molecular insights on important aspects of RAS biology and offer a resource for possible therapeutic interventions.

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

## Methods

### Recombinant expression vectors

Vectors used for recombinant expression were prepared using traditional cloning methods with custom synthesized inserts (Genewiz).

### Recombinant protein production

**MRAS, biotinylated/Avi-tagged MRAS and RAS isoforms.** All recombinant RAS isoforms were expressed and purified essentially identically. RAS vector (pET15b-6×His-HRV3C-MRAS (1–178)) was transformed into BL21 star (DE3) competent *Escherichia coli* cells (Thermo Fisher Scientific) and inoculated into TB broth supplemented with 100 μg ml$^{-1}$ carbenicillin overnight at 37 °C. Starter cultures were diluted into flasks containing 1 litre TB broth with antibiotic selection and grown at 37 °C with shaking until an optical density at 600 nm (OD$_{600}$) of around 1.0. The temperature was reduced to 17 °C, allowed to cool for around 45 min and the cultures were then induced with 0.5 mM final concentration IPTG overnight (18 h) before collecting and flash-freezing the bacterial pellets.

All of the steps were performed at 4 °C. Bacterial pellets were thawed and resuspended in lysis buffer (50 mM HEPES pH 7.5, 500 mM NaCl, 5 mM imidazole, 1 mM TCEP, 1 mM MgCl$_2$) supplemented with protease inhibitor tablets (Millipore Sigma, 11697498001, 1 tablet per 40 ml), DNase (10 μg ml$^{-1}$), and 100 mM GTP or GDP. Homogenously resuspended lysate was disrupted in a microfluidizer (Microfluidics M110L) at ~15,000 p.s.i. twice and clarified at 16,000 rpm for 1 h in a JLA 16.250 rotor (Beckman). A pre-equilibrated column (lysis buffer) of Ni-NTA resin (Qiagen, 30450, 1 ml resin bed per litre expressed) was prepared and the clarified lysate was batch-bound for 1–2 h before being drained by gravity. Beads were washed thoroughly with lysis buffer and eluted with 30 ml elution buffer (lysis buffer supplemented with 400 mM imidazole). Eluted protein was cleaved overnight by His HRV3C protease (custom preparation, 1:50 molar ratio). Cleavage was confirmed by liquid chromatography coupled with mass spectrometry (LC–MS) and the solution was passed over another Ni-NTA column pre-equilibrated with lysis buffer, collecting the flow-through. The flow-through was pooled and concentrated before loading onto a Superdex 75 HiLoad 26/600 size-exclusion column (Cytiva) in SEC Buffer (20 mM HEPES pH 7.5, 150 mM NaCl, 40 μM GDP, 5 mM MgCl$_2$). Eluting protein was evaluated by SDS–PAGE and LC–MS, pooled and frozen at around 20 mg ml$^{-1}$.

Avi-tagged versions were prepared by co-transformation of RAS (pET15b-6×His-HRV3C-RAS-Avi; MRAS 1–178, NRAS/KRAS 1–168) and BirA (pACYC184-BirA) vectors into BL21 star (DE3) competent cells, according to the RAS expression protocol with the addition of 34 μg ml$^{-1}$ chloramphenicol to the medium for selection of BirA. In vitro biotinylation was performed by the addition of biotin (100 mM final concentration) and ATP (1 mM final concentration) during cell pellet resuspension to the above protocol, retaining all of the steps described above.

**RAS nucleotide exchange.** GppNHp-bound RAS isoforms were generated by exchanging GDP-bound versions purified above with non-hydrolysable GTP analogues. In brief, EDTA (25 mM final) and 24-fold molar excess GppNHp (Jena Biosciences, 100 mM solution in 0.1 M Tris pH 8) were added to 0.5–1 μmol of purified RAS to obtain a final volume of 2.5 ml, and incubated 1 h at room temperature. PD-10 size-exclusion gravity columns (Cytiva) were equilibrated with nucleotide-exchange buffer (40 mM Tris pH 8, 200 mM (NH$_4$)$_2$SO$_4$, 0.1 mM ZnCl$_2$) according to the manufacturer's instructions. The RAS sample was loaded onto the column and eluted with 3.5 ml of nucleotide exchange buffer. A fresh 24-fold excess of GppNHp was added, along with 50 μl of shrimp alkaline phosphatase (rSAP, NEB, M0371S). The solution was incubated for 1 h at 4 °C. To quench, 210 μl of 1 M MgCl$_2$ was added to a final concentration of 30 mM. The solution was then diluted with SEC buffer (20 mM HEPES pH 7.5, 150 mM NaCl, 10 μM GppNHp, 5 mM MgCl$_2$) to about 10 ml, before final purification on a Superdex 75 HiLoad 26/600 size-exclusion column. Pure fractions were pooled, concentrated and frozen at 10–20 mg ml$^{-1}$.

**SHOC2.** SHOC2 and the M173I point mutant were prepared identically. *SHOC2* vector (pFastBac-10xHis-SHOC2) was transformed into DH10α competent *E. coli* cells and bacmid DNA was isolated according to the Bac-to-Bac Baculovirus expression system (Gibco). Bacmid DNA was transfected into a SF9 insect cell culture according to the user guidelines and high-titre P2 virus stocks were produced as described. Protein expression was carried out in SF21 insect cell culture by addition of 10 ml l$^{-1}$ P2 viral solution to log-phase SF21 cells in suspension. After 3 days, cells were collected and flash-frozen. SF9 and SF21 cells were obtained from expression systems and were not tested for mycoplasma.

All of the purification steps were performed at 4 °C. Frozen cell paste was thawed and resuspended in lysis buffer (50 mM Tris pH 8, 300 mM NaCl, 10% glycerol, 20 mM imidazole, 1 mM TCEP) supplemented with protease tablets, 2 mM MgCl$_2$ and 200 μg ml$^{-1}$ DNase. Homogenously resuspended cells were lysed twice in a microfluidizer at around 15,000 p.s.i. and the lysate was clarified for 1 h at 16,000 rpm in a JLA 16.250 rotor. A pre-equilibrated column of Ni-NTA resin (Qiagen, 30450, 1 ml resin bed per litre expressed) was prepared and clarified lysate was batch-bound for 1–2 h before being drained by gravity. The beads were washed thoroughly with lysis buffer, and eluted with 30 ml elution buffer (lysis buffer supplemented with 300 mM imidazole). Eluted protein was cleaved overnight by HRV3C protease (1:50 molar ratio) while dialysing into dialysis buffer (50 mM Tris pH 8, 300 mM NaCl, 10% glycerol, 1 mM TCEP). After LC–MS confirmation of cleavage, the solution was diluted to 50 mM NaCl with dilution buffer (50 mM Tris pH 8, 10% glycerol, 1 mM TCEP), and applied to a pre-equilibrated 5 ml HiTrap SP cation-exchange column using a sample pump. Protein was eluted through a twenty-column volume gradient between 50 mM and 1 M NaCl, evaluated by SDS–PAGE and pooled. The sample was concentrated to around 12 ml and injected into a Superdex 75 HiLoad 26/600 for final SEC purification in SEC buffer (50 mM Tris pH 7, 150 mM NaCl, 1 mM TCEP). Pure fractions were identified by SDS–PAGE, pooled, concentrated and flash-frozen at 5–15 mg ml$^{-1}$.

**PP1Cα.** PP1Cα (7–300) was produced based on previously published protocols[27,40]. In brief, PP1Cα (pET15b-6×His-PP1Cα) was co-transformed with pGRO7 plasmid encoding GroEL/GroES (Takara Bio). Transformants were grown in an LB starter culture with 100 μg ml$^{-1}$ carbenicillin, 34 μg ml$^{-1}$ chloramphenicol and 1 mM MnCl$_2$. This was used to inoculate larger 1 l cultures with the same supplements and grown until an OD$_{600}$ of ~0.5. At this point, arabinose (2 g l$^{-1}$ final concentration) was added to induce chaperone production and the cultures were grown until an OD$_{600}$ of ~1.0, after which 0.1 mM final IPTG was added to induce protein production (overnight at 10 °C). Cultures were then independently centrifuged to collect pellets and resuspended in fresh LB with 200 μg ml$^{-1}$ chloramphenicol (to prevent further ribosomal synthesis) and 1 mM MnCl$_2$. These resuspensions were shaken at 10 °C for 2 h to allow for in vivo refolding, and cell pellets were then collected by centrifugation and flash-frozen.

All of the purification steps were performed at 4 °C. Frozen cell paste was thawed and resuspended in lysis buffer (25 mM Tris pH 7, 700 mM NaCl, 5 mM imidazole, 1 mM MnCl$_2$, 0.1% Triton X-100) freshly supplemented with protease tablets. Homogenously resuspended cells were lysed twice in a microfluidizer at around 15,000 p.s.i. and the lysate was clarified for 1 h at 16,000 rpm in a JLA 16.250 rotor. A lysis-buffer-pre-equilibrated column of Ni-NTA resin (Qiagen, 30450, 1 ml resin bed per litre expressed) was prepared and clarified lysate was batch-bound for 1–2 h before being drained by gravity. The beads were washed thoroughly with consecutive PP1Cα buffer A (25 mM Tris pH 7, 700 mM NaCl, 5 mM imidazole, 1 mM MnCl$_2$) and PP1Cα wash buffer (25 mM Tris pH 7, 700 mM NaCl, 15 mM imidazole, 1 mM MnCl$_2$), and eluted with 15 ml elution buffer (25 mM Tris pH 7, 700 mM

NaCl, 250 mM imidazole, 1 mM MnCl$_2$). Eluate was applied directly to a pre-equilibrated Superdex 75 HiLoad 26/600 for initial purification in PP1Cα buffer A. Fractions were evaluated for PP1Cα by SDS–PAGE, pooled and His-TEV protease was added to cleave purification tags (1:20 molar ratio, produced in house) overnight. We consistently observed precipitation at this step, which did not prevent us from obtaining a pure final product. The cleaved solution was confirmed by LC–MS and then applied to a fresh buffer-A-equilibrated Ni-NTA column from which the flow-through was collected. The flow-through was concentrated to 15 ml and injected again into a pre-equilibrated Superdex 75 HiLoad 26/600 for final purification into either SEC buffer (50 mM Bis-Tris pH 6.5, 500 mM NaCl, 1 mM TCEP) or directly into SPR running buffer (20 mM Bis-Tris pH 6.5, 200 mM NaCl, 2.5% glycerol, 1 mM TCEP, 0.5 mM MgCl$_2$, 5 μM GppNHp, 0.005% Tween-20). The fractions were pooled, concentrated and used fresh for downstream binding studies and ternary complex formation.

**cRaf1 RBD with and without an N-terminal His tag.** The *E. coli* expression construct used in this study was based on the pET system and was generated using standard molecular cloning techniques. The DNA encoding the cRAF RBD (amino acids 51–131 of the full-length protein) was inserted after a cleavable N-terminal hexa histidine-tag. The insert was codon-optimized and synthesized by GeneArt (Thermo Fisher Scientific). The final expression construct was verified by Sanger sequencing.

Two litres of culture medium were inoculated with a pre-culture of *E. coli* BL21(DE3) freshly transformed with the expression plasmid, and protein expression was induced with 1 mM IPTG (Sigma-Aldrich) for 16 h at 18 °C.

The bacterial cells were collected by centrifugation and the pellets were frozen on dry ice. The cell pellets were then resuspended in buffer A (50 mM HEPES, 300 mM NaCl, 20 mM imidazole, 1 mM TCEP, pH 7.8) supplemented with Turbonuclease (Merck) and complete protease inhibitor tablets (Roche). The cells were lysed by three passages through a homogenizer (Avestin) at about 1,000 bar and the lysate was clarified by centrifugation at 40,000*g* for 40 min.

The lysate was loaded onto the HiFliQ 5 ml Ni Advance column (ProteinArk), mounted on an ÄKTA Pure 25 chromatography system (Cytiva). Contaminating proteins were washed away with buffer A and bound protein was eluted with a linear gradient to buffer B (buffer A supplemented with 400 mM imidazole).

Constructs without an N-terminal His tag were obtained by HRV 3C protease cleavage during dialysis overnight against buffer A. The protein solution was reloaded onto the Ni Advance column and the flow through containing the target protein was collected.

Both tagged and untagged proteins were then further purified over a HiLoad 26/600 Superdex 75 pg column (Cytiva) pre-equilibrated with storage buffer (50 mM HEPES, 300 mM NaCl, 1 mM TCEP, 10% glycerol, pH 7.4).

The purity and concentration of the protein was determined by reversed-phase high-performance liquid chromatography, and its identity was confirmed by LC–MS.

## SPR binding studies
**Binary MRAS–SHOC2 and ternary complex SPR.** Ternary complex binding studies were carried out on a Biacore 8K SPR instrument (Cytiva) at 25 °C using modified buffer for PP1Cα stability (20 mM Bis-Tris pH 6.5, 200 mM NaCl, 2.5% glycerol, 0.5 mM MgCl$_2$, 1 mM TCEP, 5 μM GppNHp or GTP, 0.005% Tween-20). Biotinylated MRAS–avi, NRAS–avi, KRAS–avi or SHOC2–avi protein was immobilized onto series SA streptavidin chips at a flow rate of 5 μl min$^{-1}$ to a surface density of 500 RU. Parallel kinetics runs were used to evaluate seven-point dose–response curves simultaneously with 90 s association time and 600 s dissociation time at a flow rate of 20 μl min$^{-1}$. PP1Cα was not observed to interact with either MRAS or SHOC2 independently and was held in constant excess of the analyte for dose–response curves.

Data were analysed for steady-state affinity using the Biacore evaluation software.

MRAS nucleotide loading state binding experiments (Extended Data Fig. 1a only) were carried out on a Biacore T200 SPR instrument (Cytiva) at 25 °C using standard running buffer (20 mM HEPES pH 7.5, 150 mM NaCl, 0.5 mM MgCl$_2$, 1 mM TCEP, 5 μM GppNHp or GDP, 0.005% Tween-20). Biotinylated MRAS (500 RU) was immobilized onto a series S SA chip. Multi-cycle runs were performed with a 60 s association time and a 1,200 s dissociation time.

**Binary RAS-RBD SPR.** Binding studies were performed on the Biacore 8K SPR instrument (Cytiva) at 25 °C. Biotinylated WT and mutant MRAS–avi proteins were diluted to 0.04 μg ml$^{-1}$ in assay buffer (50 mM HEPES pH 7.5, 150 mM NaCl, 1 mM MgCl$_2$, 10 μM GppNHp, 0.1% Tween-20) and immobilized on a SA streptavidin chip at a flow rate of 5 μl min$^{-1}$ to a surface density of about 100 RUs. Raf1 RBD (residues 51–131) was injected in multi-cycle runs up to 4 μM, at a flow rate of 50 μl min$^{-1}$ over all flow cells with 90 s association and 100 s dissociation times. Data were analysed using the Biacore evaluation software using a 1:1 binding model.

## Sedimentation velocity analytical ultracentrifugation
Sedimentation velocity experiments were performed in the Beckman-Coulter Optima AUC analytical ultracentrifuge. Samples were dialysed into analytical ultracentrifugation buffer (20 mM Bis-Tris pH 6.5, 500 mM NaCl, 0.5 mM MgCl$_2$, 1 mM TCEP, 5 μM GppNHp) and were loaded into dual-sector centrifuge cells equipped with 1.2 cm centrepieces and sapphire windows. The cells were loaded into an AN-50 Ti rotor and allowed to equilibrate to 20 °C for 2 h before the start of high-speed sedimentation at 42,000 rpm. Data were collected at 280 nm and examined using $c(s)$ analysis[41] (SEDFIT v.5.01b). Results in the form of a $c(s)$ plot were visualized with Gussi[42].

## Semi-preparative SEC and MRAS/KRAS competition
Ternary complex was formed and evaluated using SEC on a Superdex 200 10/300 GL column (Cytiva). MRAS(1–178)–GppNHp, SHOC2(M173I, 80–582) and PP1Cα(7–300) were combined simultaneously into a ~500 μl reaction at a final concentration of 30–50 μM. To avoid SHOC2 or PP1Cα monomers (which cannot be fully separated from ternary complex), 1.2-fold molar ratios were used for MRAS. After 10 min, the mixture was dialysed against SEC buffer (20 mM HEPES pH 7.5, 150 mM NaCl, 1 mM MgCl$_2$, 1 mM TCEP) for 2–4 h at 4 °C. Dialysed solution was applied to the Superdex 200 10/300 column and a strong ternary complex (and weak monomeric RAS) peak was observed. Fractions corresponding to the ternary complex were assessed by SDS–PAGE, pooled and concentrated to ~6.5 mg ml$^{-1}$ for crystallization trials.

In the competition SEC experiment with MRAS and KRAS, SHOC2, PP1Cα, MRAS(1–178) and KRAS(1–169) were combined at a 1:1:1.2:1.2 molar ratio. To avoid concerns with the order of addition, MRAS–GppNHp and KRAS–GppNHp were first combined together before adding to a mixture of SHOC2(M173I, 80–582) and PP1Cα(7–300). After around 2 h incubation at 4 °C, the mixture was applied to a Superdex 200 10/300 GL column (Cytiva) that had been equilibrated with SEC buffer (20 mM HEPES pH 7.5, 150 mM NaCl, 1 mM MgCl$_2$, 1 mM TCEP, 5 μM GppNHp). The ternary complex of SHOC2–KRAS–PP1Cα was prepared in the same manner but without MRAS. Fractions across the elution profile were analysed by SDS–PAGE with Coomassie staining.

## Crystallization
The sitting-drop vapour-diffusion method was used for crystallization of both SHOC2(80–582) apo and SHOC2(80–582) in complex with MRAS(1–178) and PP1Cα(7–300). ApoSHOC2 was crystallized using a 1:1 volume of protein and well solution (0.1 M Tris-Cl, pH 8.0, 0.2 M MgCl$_2$, 11% PEG 4000, 2.5% 1,5-diaminopentane). Crystals were cryoprotected using the crystallization solution with the addition of

20% glycerol followed by flash-freezing directly into liquid nitrogen. The ternary complex was co-crystallized using a 1:1 volume of protein and well solution (0.1 M NaH$_2$PO$_4$ pH 6.5, 12% PEG 8000). Crystals were cryoprotected using the crystallization solution with the addition of 20% glycerol followed by flash-freezing directly into liquid nitrogen.

## Structure determination

Diffraction data for both apoSHOC2 and the ternary complex were collected on the Dectris Pilatus 6M Detector at beamline 17ID (IMCA-CAT) at the Advanced Photon Source at Argonne National Laboratories. The data were measured from a single crystal maintained at 100 K and exposed to a wavelength of 1 Å, and the reflections were indexed, integrated and scaled using XDS[35]. The space group of apoSHOC2 was C121 with two molecules in the asymmetric unit. The structure was determined by molecular replacement using Phaser[43], using *Leptospira interrogans* LRR protein LIC12234 (PDB: 4TZH) as a search model. The final model was built in the Coot molecular graphics application[44] and refined in Phenix[45]. The space group of the ternary complex was P1211, with two complexes in the asymmetric unit. The structure was determined by molecular replacement, using the protein phosphatase 1 (PP1) T320E mutant (PDB: 6ZK6), MRAS in complex with GppNHp (PDB: 1X1S) and apo SHOC2 as search models with all solvent molecules removed. The final model was built in the Coot molecular graphics application and refined using Buster[46] and Phenix. The final model had geometry within the acceptable limits (Extended Data Table 1) with 0.05% of residues in Ramchandran disallowed regions. Data collection and refinement statistics are shown in Supplementary Table 1.

## Molecular dynamics simulations and modelling

**Molecular dynamics simulations.** The starting structure for modelling efforts was the solved ternary complex of GppNHp-bound MRAS(Q71R), SHOC2(M173I) and PP1Cα with Mn, phosphate ions in the PP1Cα active site. Other cofactors were removed, and high-occupancy crystal waters were retained for subsequent processing in Schrodinger[47] through the protein preparation feature, followed by manual curation of rotameric and protonation states for consistency of the hydrogen bonding network. Free ends were capped with neutral ACE and NME residues. To model the WT states of both MRAS and SHOC2, residues at MRAS position 71 and SHOC2 position 173 were mutated back to their corresponding WT counterparts. For both mutant and WT versions of the ternary complex, supplemental waters were placed using 3D-RISM[48,49]. Proteins were modelled using the AMBER ff19SB protein force field[50], GppNHp was parameterized via the parm@Frosst force field[51] and water was modelled as TIP3P[52].

Molecular dynamics simulations were carried out using AMBER 16[53] and PMEMD CUDA[54–56] starting from both the mutant and WT (modelled) structures. Each system was first minimized (keeping solute molecules fixed) starting with 10,000 steps of steepest descent followed by 10,000 steps of conjugate gradient minimization using constant-volume periodic boundaries. The system was heated to 100 K over 2,500 steps (5 ps) using Langevin dynamics[57] with a constant volume. At a constant pressure, the system was then warmed from 100 K to 303 K over a period of 100 ps, using a time step of 2 fs. Minimization and heating steps were run with fixed solute molecules using a force constant of 5.0 kcal mol$^{-1}$ Å$^2$. Next, the system was run at 303 K for 100 ps, in which all Cα atoms, GppNHp, Mg, Mn, phosphate ions were restrained by a force constant of 10.0 kcal mol$^{-1}$ Å$^2$. Three replicates of final 1 μs production runs at a constant pressure and temperature (NPT) were submitted.

**Clustering simulation data.** The first 1,000 frames (50 ns) were removed from each trajectory to minimize the inclusion of insufficiently equilibrated frames in the analysis. The remaining 19,000 frames were pooled from each of the three trajectory replicates, and clustered using CPPTRAJ[58] on the basis of interface residues between SHOC2 and MRAS about MRAS residue 71 and SHOC2 residue 173. Clustering was performed using the *k*-means algorithm and the optimal number of clusters, *n* = 6, was determined using the elbow method. An average structure for each replicate was generated using CPPTRAJ, and representatives were pulled for each cluster as examples of key conformations of both the WT and mutant simulations. Analysis of protein–protein and protein–water interactions was performed using a combination of CPPTRAJ scripts and the getcontacts software package[59].

## TR-FRET assay to assess competition of Ras/cRaf1-RBD binding

A panel of TR-FRET assays scoring for interaction of different Ras constructs (amino acids 1–168 for NRAS, 1–178 for MRAS constructs) with cRaf1 RBD (amino acids 51–131, RBD) were set up. The final concentrations in the assay were 25 nM C-terminally avi-tagged and biotinylated RAS, 6.25 nM Cy5–streptavidin (VWR), 50 nM RBD and 1 nM anti-6×His Europium (LANCE Eu-W1024 Anti-6×His, Perkin Elmer) in assay buffer (50 mM Tris pH 7.5, 100 mM NaCl, 1 mM MgCl$_2$, 50 μM GppNHp, 0.5 mM DTT, 0.005% Tween-20).

RAS constructs were pre-incubated for 30 min with Cy5–streptavidin (VWR) in assay buffer at 2.2× final concentration. In parallel, His–RBD (51–131) was pre-incubated for 30 min with anti-6×His Europium at 2.2× final concentration. Twofold serial dilutions of untagged protein competitors were added to the RAS/Cy5–streptavidin mix and incubated for 30 min before adding the RBD/anti-His Europium mixture.

After an additional 1 h incubation at room temperature, the plates were measured on the PHERAstar FSX plate reader (BMG Labtech) with excitation at 320 nm and emission at 620 nm and 665 nm wavelengths. TR-FRET ratios were calculated by dividing counts at 665 nm by counts at 620 nm and multiplying by 100. Data were normalized to percentage activity and plotted in GraphPad Prism (v.9.2.0, GraphPad; www.graphpad.com). Curves were fitted using the equations for non-linear regression, log[inhibitor] versus response, variable slope (four parameters), fixing the lower plateau to 0.

## M2H assay

The NanoLuciferase reporter cell line (HEK293A-M2H-NLuc, tested negative for mycoplasma contamination) was generated by lentiviral transduction of a custom vector (pNGX_LV_017_GAL4_NLuc) bearing 5×GAL4 motifs upstream of a minimal promoter and the NanoLuciferase open reading frame (ORF). GAL4-DBD or VP16-TAD ORFs were cloned into custom lentiviral vectors (named pXP1510 and pXP1512) under an *EF1a* promoter and in frame with a ccdb cassette flanked by type IIS enzymes to enable golden-gate cloning of *SHOC2* or *MRAS* cDNA (obtained by gene synthesis). Mutants were obtained by site-directed mutagenesis (Agilent QuickChange Lightning kit). HEK293A-M2H-Nluc cells were maintained in DMEM (Gibco) supplemented with 10% FBS (Gibco). Cells were plated at $1.25 \times 10^4$ cells per well in a white transparent-bottom 96-well plates and transfected the next day with combinations of GAL4–MRAS and VP16–SHOC2 constructs using X-tremeGENE 9 (Roche) at a ratio 1:3 (DNA:transfection reagent). Then, 72 h after transfection, Nano-Glo cell reagent (Promega) was added and luminescence was measured using the Synergy HT reader (BioTek). The drawing in Fig. 3c illustrating the set-up of the M2H assay was generated using Biorender (www.biorender.com).

## SHOC2–eGFP flow cytometry and imaging

All SHOC2–eGFP experiments were performed in HEK293T cells maintained in DMEM (Gibco) supplemented with 10% FBS (Gibco) and confirmed negative for mycoplasma contamination. SHOC2-eGFP-chy-mCherry constructs were custom synthesized (Vectorbuilder); the bicistronic SHOC2-eGFP-chy-mCherry cassette is expressed through a *hPGK1* promoter, and the self-cleaving chysel (chy) peptide sequence between SHOC2-eGFP and mCherry enables 1:1 expression of both proteins and enables monitoring of SHOC2-eGFP turnover on the basis of eGFP/mCherry ratio measurements[60] using

flow cytometry or imaging for SHOC2 localization studies. For both, 16,000 HEK293T cells were transfected with 120 ng of plasmid in 96-well format 1 day after seeding using the FuGene6 (Promega) standard protocol. Then, 48 h after transfection, cells were prepared from 96-well standard tissue culture plates for flow cytometry, data were acquired with a CytoFLEX S instrument (Beckman Coulter) and analysed using FlowJo v.10.7.1 software. Imaging studies for SHOC2 localization were performed with an InCell analyser6500 (GE Healthcare) from 96-well cell-culture-treated black-wall chambers (Ibidi), using a Nikon 40×/0.95 Plan Apo, Corr Collar 0.11–0.23, CFI/60 lambda objective and the following conditions and filter-set (green: Exc 488/10, Emi 524/48, 0.2 s exposure, 2 × 2 binning; orange: Exc 561/1, Emi 605/51, 0.2 s exposure, 2 × 2 binning).

## MiaPaca2 cell line generation
All MiaPaca2 cell lines were grown in DMEM (Gibco) supplemented with 10% FCS (Bioconcept), 1% sodium pyruvate (Bioconcept) and 1% glutamine (Bioconcept). MiaPaca22 *SHOC2*-KO cells were generated in MiaPaca2 cells stably expressing *Cas9* (MiaPaca2 *Cas9*) generated by lentiviral transduction of a custom vector expressing sp*Cas9* under an *EF1A* promoter (named pNGX_LV_c028). MiaPaca2 lines tested negative for mycoplasma contamination. In summary, 30,000 MiaPaca2 CAS9 cells were seeded in 6-well plates and, the next day, were transfected with 2 µM SHOC2 sgRNA (5′-TAGTTATACGATTAAAGCGA-3′ (18)) using 6 µl Dharmafect (Dharmacon) according to the manufacturer's manual. Then, 6 days after transfection, cells were prepared for clonal selection from the transfected pool. Clones were screened for SHOC2 levels by western blotting with an anti-SHOC2 antibody (Cell Signaling Technology, 53600) and successful KO clones were validated by sequencing, one validated *SHOC2*-KO clone (MiaPaca2 *SHOC2*-KO) was subsequently used for re-expression of various SHOC2 mutants using lentiviral transduction. For lentivirus production, HEK293FT cells were seeded in T175 cell culture flasks and transfected with 12 µg pVPR-Gag-Pol, 4.8 µg pMD2-VSV-G and 1 µg of a custom lentiviral vector pXP1510 expressing various SHOC2 mutants (pXP1510-SHOC2 mutants) using Mirus TransIT (Mirus, MIR2700). The virus-containing supernatant was collected and filtered using 0.45 µm syringe filters 48 h later and subsequently used for MiaPaCa-2 *SHOC2*-KO transduction. Then, 24 h after infection, cells were selected with neomycin (1 mg ml$^{-1}$, Thermo Fisher Scientific, 10131027).

## Western blotting
Whole-cell lysates were prepared in RIPA buffer (Thermo Fisher Scientific, 89901) supplemented with protease and phosphatase inhibitor cocktail (Sigma-Aldrich, PPC1010) and quantified using the BCA Protein Assay Kit (Thermo Fisher Scientific, 23225). Samples were loaded on 4–12% Bis-Tris gels (Invitrogen, WG1403A) and transferred to nitrocellulose membranes (Bio-Rad, 1704159) using the Trans-Blot Turbo system (Bio-Rad, 1704150). All membranes were blocked with 5% bovine serum albumin (Sigma-Aldrich, A2153-100G) and incubated with primary antibodies overnight. Immunoblots were imaged using the FUSION FX7 Imaging System (Vilber) and densitometry analysis was performed using its integrated quantification software.

## Analysis of dependency map data
We used the DepMap package (v.1.8) and experiment hub package (v.2.2), available on Bioconductor, to access the datasets from the Broad Institute DepMap cancer dependency study[61,62]. Datasets from the Q4 2021 release were used, specifically, the CRISPR–Cas9 gene KO screens quantifying the genetic dependency of 1,052 cancer cell lines (dataset ID EH7290 in experiment hub). Additional datasets available in the package pertaining to the metadata and mutation calls were used (experiment hub IDs EH7294 and EH7293, respectively). The dependency scores of genes in the cell lines are the features of primary interest. These can be interpreted as an expression of how vital a particular gene

is for a given cancer cell line. For example, a highly negative dependency score is derived from a large negative log-transformed fold change in the population of cancer cells after gene KO, implying that a given cell line is highly dependent on that gene.

To investigate the relationship between the knockdown of a *RAS* gene and of *SHOC2*, we calculated the Pearson correlation of the dependency scores using the stats library core R package[63]. Specifically, we calculated the Pearson correlation between the dependency score of *SHOC2* and the dependency score of each *RAS* gene (*MRAS*, *KRAS*, *NRAS*, *HRAS*) in each group of cell lines which contained the mutation of interest (p.G12-, p.G13-, p.Q61-, other, WT). We did not distinguish between what the amino acid changed to, for example, between p.G12C, p.G12V). This enabled us to investigate the linear relationship of the two genes in each of the mutation groups. The significance of the correlation in each group was assessed. A *P* value on the *t*-test of each correlation (testing the alternative hypothesis that the population correlation is significantly greater than 0) was calculated. All analysis was performed using R v.4.1.1.

## Recombinant BRAF dephosphorylation assay
Purified autoinhibited and active BRAF–MEK1–14-3-3 complexes were prepared as previously described[4]. BRAF complexes (at a concentration of 500 nM BRAF) were incubated with an equimolar ratio of the WT or mutant MRAS–SHOC2–PP1Cα complex, KRAS–SHOC2–PP1Cα complex, PP1Cα alone, or lambda phosphatase for 90 min at 30 °C in a buffer containing 100 mM HEPES (pH 7.4), 150 mM NaCl, 2 mM MnCl, 1 mM TCEP and 0.02% Brij 35. Reaction products were analysed using western blotting using phosphospecific antibodies for the BRAF phosphorylated at Ser365 (Cell Signaling Technology, 9421S) or Ser729 (Abcam, ab124794). Aliquots of each reaction were resolved on a Coomassie-stained SDS–PAGE gel to provide a loading control for the BRAF complexes.

## Reporting summary
Further information on research design is available in the Nature Research Reporting Summary linked to this article.

## Data availability
The data supporting the findings of this study are available from the corresponding authors upon reasonable request. Structural factors and coordinates have been deposited at the PDB under accession codes 7TYG for the SHOC2(80–582) apo structure and 7TXH for the SHOC2(M173I, 80–582)–MRAS(Q71R, 1–178)–PP1Cα(7–300) ternary complex structure.

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

**Acknowledgements** We thank our extended team members and J.-R. Marchand, S. Gleim, J. Koschwanez, T. Schmelzle, J. Tallarico and S. Malek for support and scientific input; the NIBR Innovation Postdoctoral program for support of Z.J.H.; B. Yang for MS analysis; and M. Meyerhofer, C. Zimmermann, P. Fontana, C. Premand, W. A. Rahman and D. Erdmann for support in protein production.

**Author contributions** Z.J.H., D.A.K., C.R.T. and L.T. planned the study, designed the experiments and analysed results. M.F., Z.J.H. and D.A.K. solved the structures with support from K.C. and A.D.; Z.J.H. and D.A.K. wrote the manuscript with help from L.T. and C.R.T.; A.D. and J.V. supported the biophysical studies performed by Z.J.H.; D.E. performed cellular in vitro studies supported by H.G. and S.M.; F.M. performed the SHOC-GFP studies. G.K. performed the DepMap analysis. M.B. performed M2H assays. S.K. performed TR-FRET. E.P. performed in vitro activity assays and D.M.J. performed SEC competition experiments, both with support from M.J.E.; K.A.P. and C.V.-V. performed the molecular dynamics simulations. K.S.B., G.G.G., S.-M.M., T.S. and M.J.E. provided intellectual input.

**Competing interests** Z.J.H., A.D., J.V., D.E., M.B., S.K., K.A.P., F.M., H.G., G.K., S.M., C.V.-V., K.S.B., G.G.G., S.-M.M., T.S., K.C., L.T., C.R.T. and D.A.K. are employees and shareholders of Novartis Pharma. M.F. is a former employee of Novartis. M.J.E. receives funding from Novartis Pharma.

**Additional information**
**Correspondence and requests for materials** should be addressed to Zachary J. Hauseman, Luca Tordella, Claudio R. Thoma or Daniel A. King.

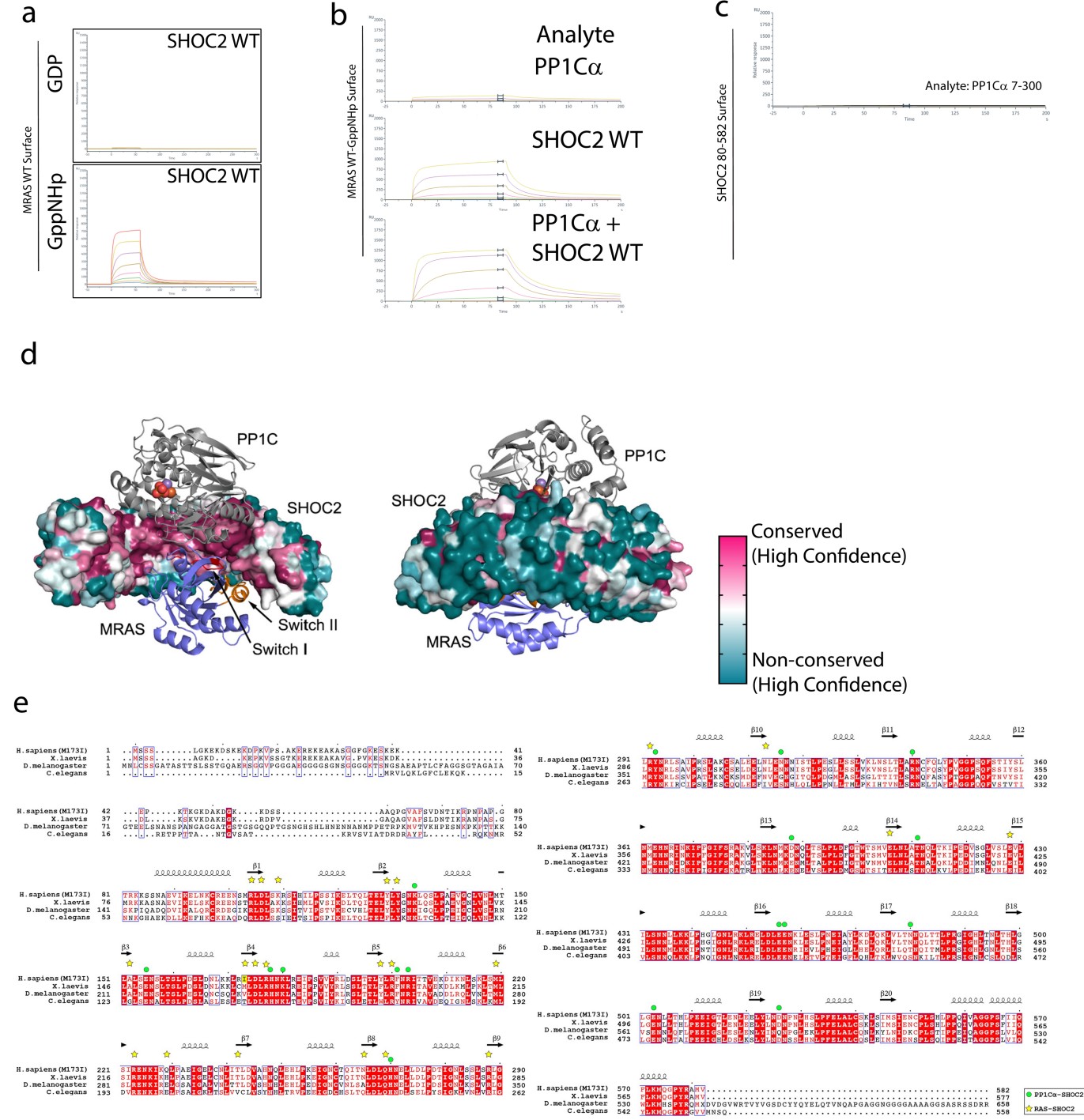

**Extended Data Fig. 1 | SHOC2 binds MRAS and PP1 through the highly conserved concave face. a**. Recombinant MRAS loaded with either GDP or non-hydrolysable GTP analogue GppNHp was immobilized on a streptavidin (SA) chip and recombinant SHOC2 80-582 was flowed over in dose response. Representative sensorgrams are shown, where only active GppNHp loaded MRAS could bind to SHOC2. **b**. Recombinant MRAS loaded with non-hydrolysable GTP analogue GppNHp was immobilized on a streptavidin (SA) chip and recombinant PP1Cα, SHOC2 80-582 or FL SHOC2 was flowed over in dose response in presence of excess PP1Cα. Representative sensorgrams are shown, where PP1Cα binds to active GppNHp loaded MRAS only on the presence of SHOC2 80-582. **c**. Recombinant SHOC2 80-582 was immobilized on a streptavidin (SA) chip and recombinant PP1Cα was flowed over in dose response. **d**. PP1C and MRAS bind a highly conserved surface on SHOC2.

Front and back views of the ternary SHOC2/MRAS/PP1C complex. Note that SHOC2 engages PP1C and MRAS with a highly conserved region on its concave surface. By contrast, the outer, convex surface of SHOC2 is poorly conserved. The surface of SHOC2 is coloured according to conservation from magenta (most conserved) to teal (most variable) as analysed with the CONSURF server (PMID: 27166375). **e**. SHOC2 sequence alignment. Amino acid sequences of human, frog, fly and worm SHOC2 are aligned and identically conserved residues are shaded red. The site of the M173I mutation in the present structure is shaded yellow. Secondary structure elements of SHOC2 are indicated above the alignment. Symbols above the alignment indicate residues that in the holoenzyme complex lie in the interface with PP1Cα (green dots) or MRAS (yellow stars).

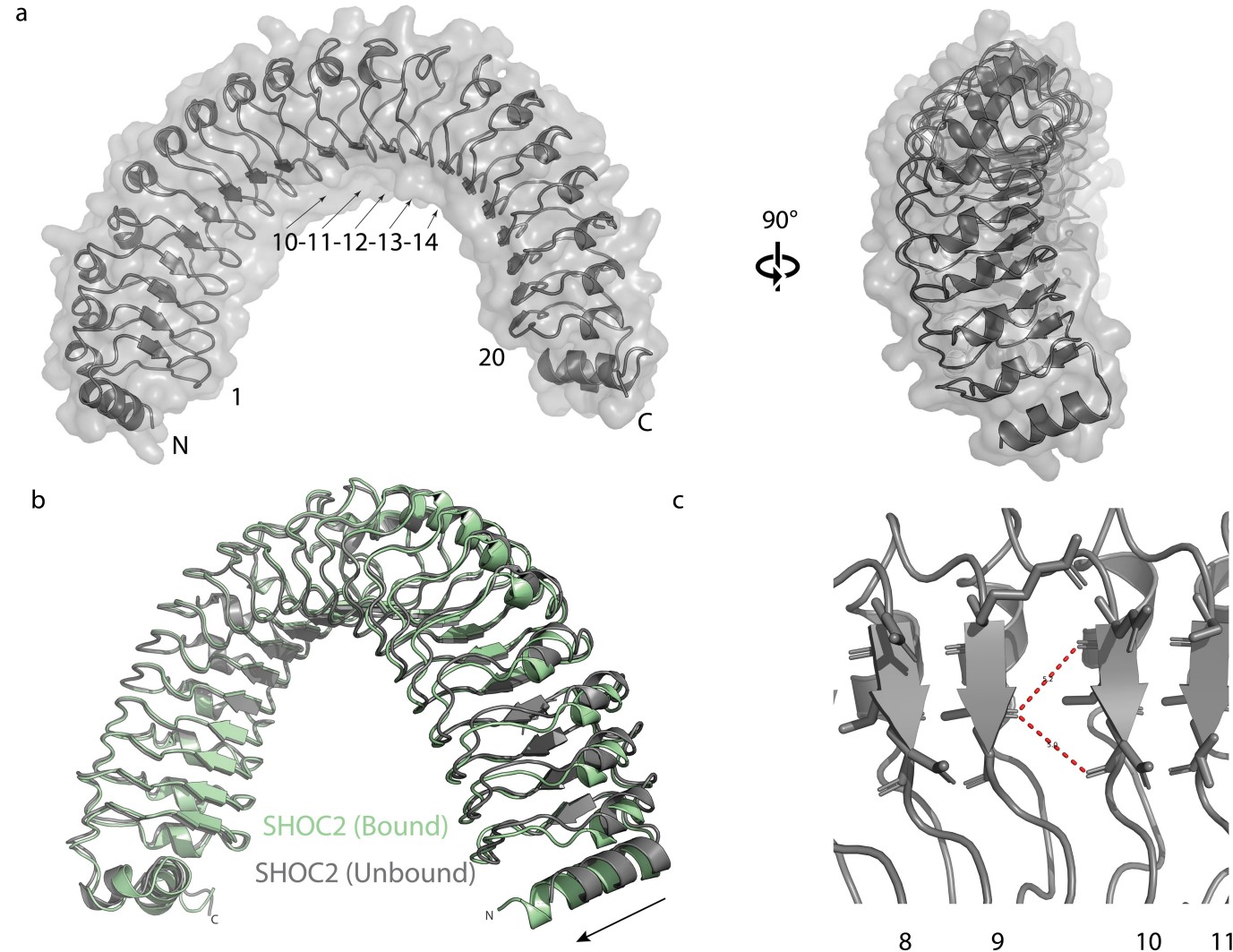

**Extended Data Fig. 2 | Crystal structure of SHOC2 80-582. a**. Crystal structure of recombinant SHOC2 80-582 reveals a folded domain of 20 leucine rich repeats in excellent agreement with structure predictions. Both N and C terminal capping motifs are resolved, without density for the portion of the unstructured N terminus retained. LRR repeats are numbered. **b**. Alignment of the SHOC2 apo structure (grey) and SHOC2 from the ternary complex structure (pale green). The SHOC2 overall fold is identical, and the curvature of the solenoid shows only minor change to accommodate binding of the other complex members. **c**. The hydrogen bond network characteristic of the beta sheet traversing the concave face is disrupted between LRR 9 and 10, which is a component of slightly atypical LRR fold in the mid-range repeats of SHOC2.

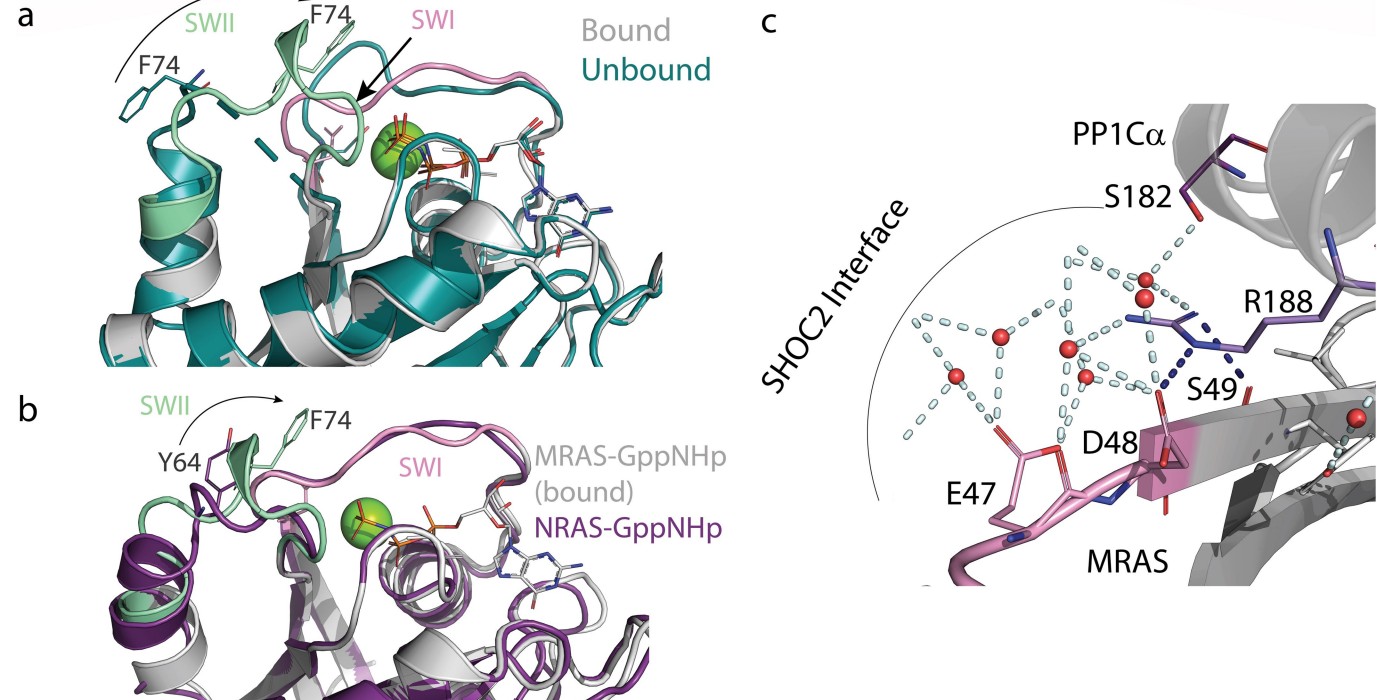

**Extended Data Fig. 3 | MRAS SWI/SWII loops adopt a closed conformation.**
**a**. MRAS shifts towards a closed conformation in the ternary complex. Zoom in of SWI and SWII region of bound MRAS-GppNHp (grey) aligned with apo MRAS-GppNHp (teal pdb: 1x1s). Movement of SWII F74 from unbound open (teal) to closed bound (pale green) is illustrated. SWI of bound MRAS is shown in pink. **b**. Bound MRAS SWI (pink) exhibits a closed conformation similar to active NRAS (purple, pdb: 5uhv). MRAS SWII (pale green) is conformationally shifted relative to NRAS-GppNHp as well. Movement of SWII F74 from NRAS counterpart Y64 is shown. MRAS-GppNHp is shown in grey. **c**. Cutout view of the water network mediated by critical residues: PP1C R188 and MRAS SWI E47, D48 and S49 mediate a coordinated network that connects all subunits of the complex.

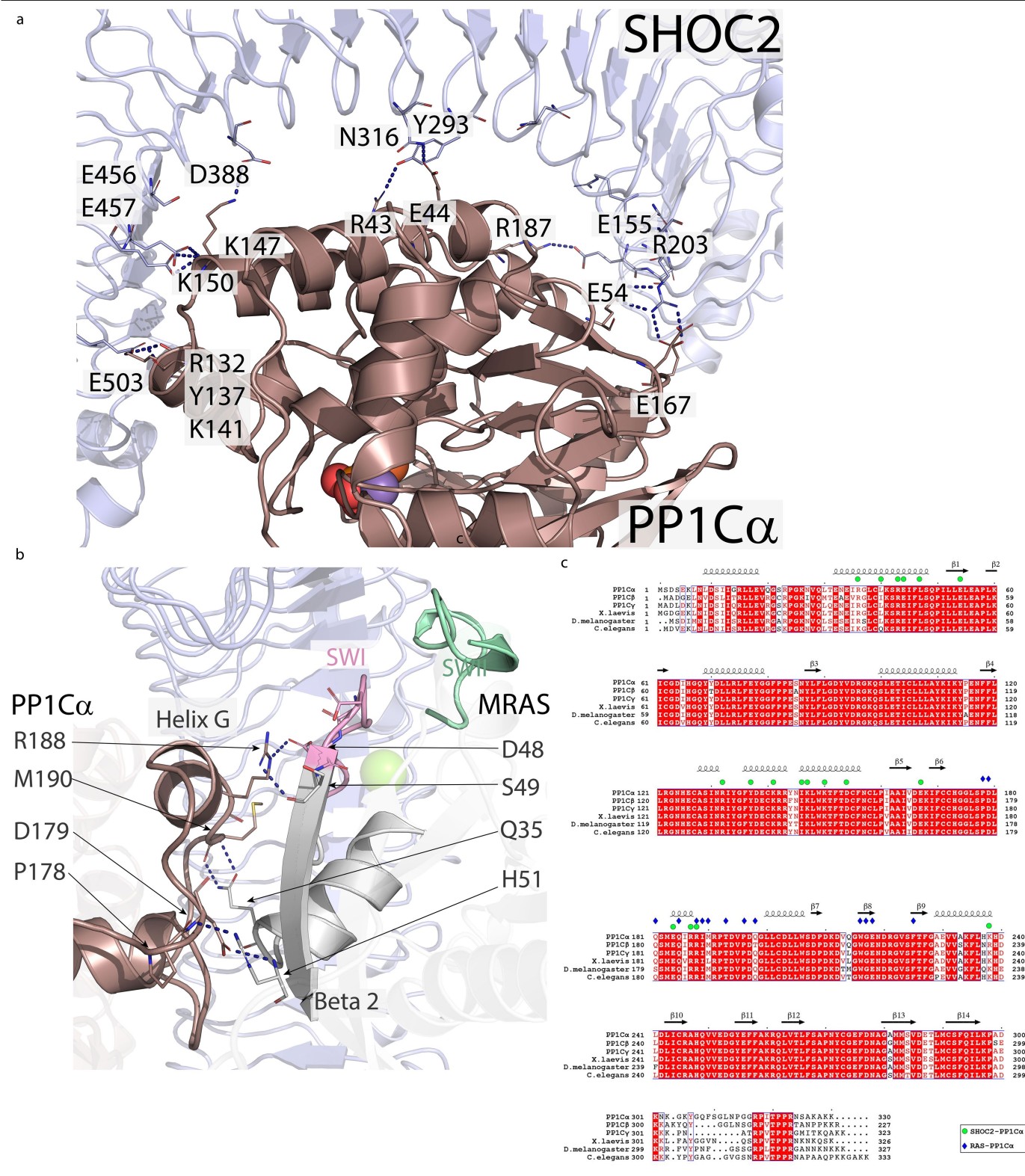

**Extended Data Fig. 4 | PP1C makes extended interactions with SHOC2 and the MRAS effector surface. a**. PP1Cα interacts with SHOC2 across an extended arc of residues that localize to the ascending loop of SHOC2 LRRs. These interactions are primarily hydrophilic and become more networked near the N-terminal SHOC2 LRRs. Selected residues with salt bridge and hydrogen bond interactions are labelled. **b**. Helix G of PP1 (purple) and SWI(pink)/Strand beta 2 (grey) are the primary interaction surfaces of the MRAS/PP1Cα PPI. Key

interacting residues are labelled. **c**. PP1C sequence alignment. Amino acid sequences of human PP1Cα, PP1Cβ, and PP1Cγ are aligned with PP1C sequences from frog, fly and worm. Identically conserved residues are shaded in red. Secondary structure elements of PP1C are indicated above the alignment. Symbols above the alignment indicate residues that in the holoenzyme complex lie in the interface with SHOC2 (green dots) or MRAS (blue diamonds).

a

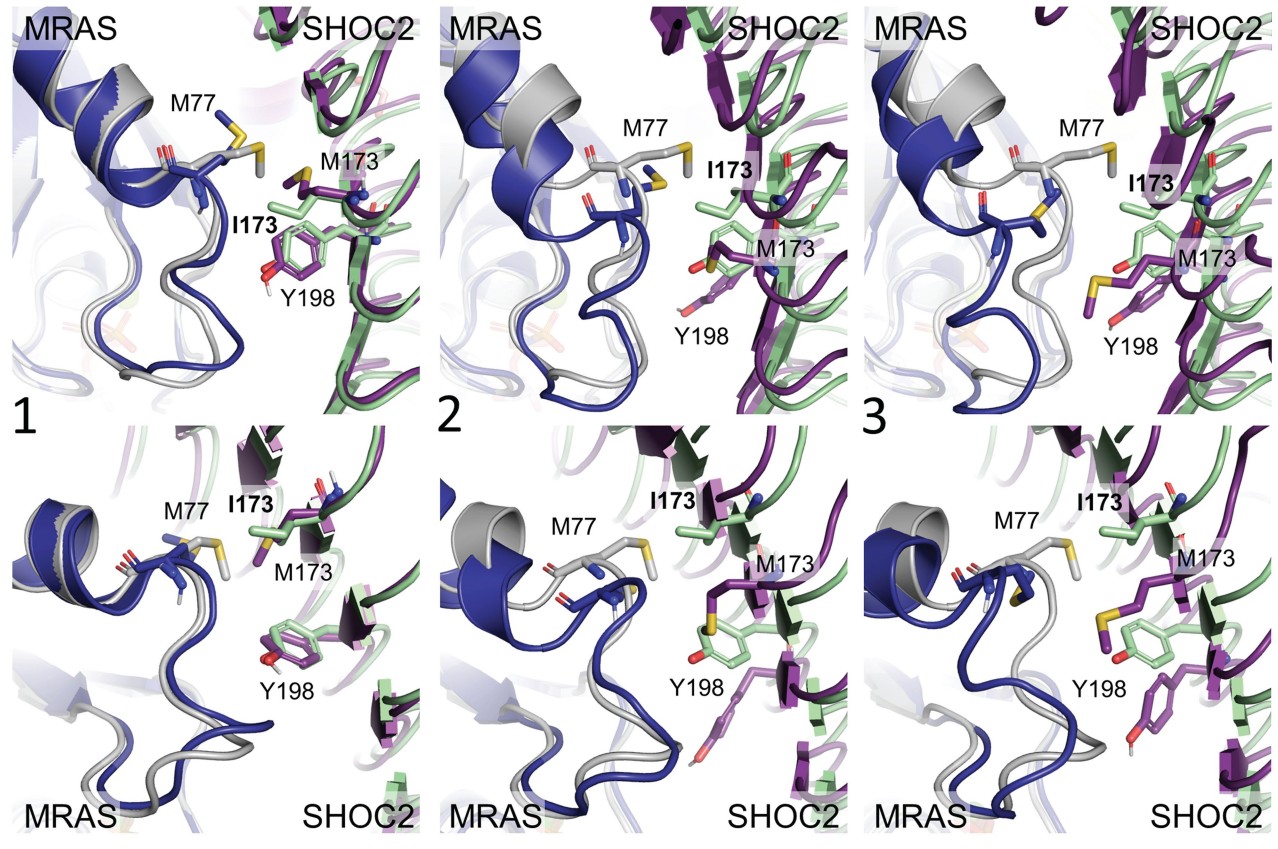

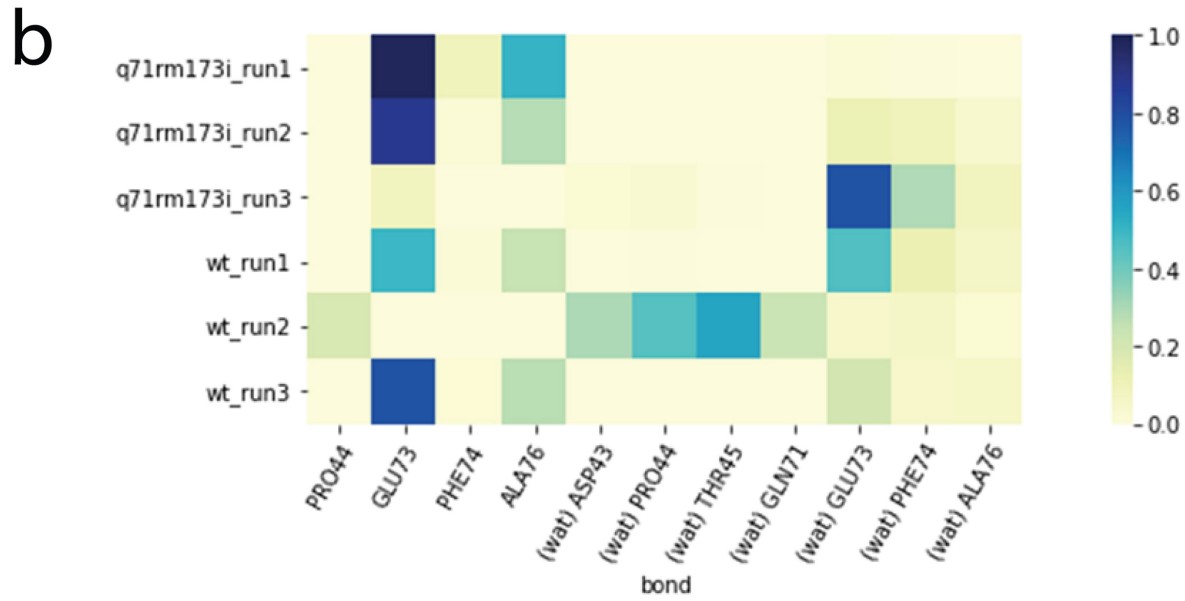

**Extended Data Fig. 5** | See next page for caption.

**Extended Data Fig. 5 | Effects of M173I mutation on the SHOC2-RAS-PP1C interface.** Three 1 μs trajectories were submitted starting from the crystal structure of SHOC2 M173I/MRAS Q71R/PP1Cα as a means of validating the crystal complex. To investigate the impact of the mutations in SHOC2 and MRAS, the WT complex was simulated by mutating SHOC2 and MRAS residues (I173, R71) to their wild type residues (M173, Q71) and submitting three additional 1 us trajectories. Frames from each of the three trajectories were pooled and clustered. Examining the cluster centres as representative structures revealed three distinct states of the MRAS WT and SHOC2 WT interface. Cluster populations approximate the frequency that each state is observed, revealing state 1 as occurring roughly 67% of the time, followed by state 2 at 29%, and state 3 at 4%. In the crystal complex and across the mutant trajectories, Y198 most frequently interacts with MRAS through hydrogen bonding to A76 and E73 backbone atoms. Y198, A76, and E73 contacts are conserved in WT runs 1 and 3. Conversely in WT run 2, in which states 2 and 3 of the MRAS WT/SHOC WT interface are most often sampled, the displacement of Y198 and MRAS switch II results in an altered hydrogen bonding network, where Y198 interfaces MRAS mostly through water-mediated interactions to D43, P44, T45, or Q71. **a.** Representative structures of the SHOC2-MRAS interface from simulated SHOC2 WT/MRAS WT/PP1Cα WT ternary complex. Three distinct states of the MRAS (dark blue) - SHOC2 (purple) interface are observed from the clustered MRAS WT/SHOC2 WT/PP1Cα WT trajectories and aligned to the crystal structure of the ternary SHOC2 M173I (pale green), MRAS Q71R (grey), PP1Cα (not shown) complex. (1) In state 1, SHOC2 M173 pushes the sidechain of MRAS M77 away from the interface, while leaving SHOC2 Y198 unperturbed. (2) In state 2, M173 is buried, resulting in the displacement of Y198, altering the hydrogen bonding network compared to that of the crystal structure. (3) In state 3, M173 directly pushes the MRAS switch II loop away from the SHOC2 interface. **b.** Frequency of observed contacts with SHOC2 Y198 across replicate MD simulations. Y198 hydrogen bonds between E73 and A76 are lost in the second SHOC2 WT/MRAS WT/PP1Cα WT trajectory. Replacement contacts between Y198 and MRAS residues are mostly water-mediated (indicated as 'wat').

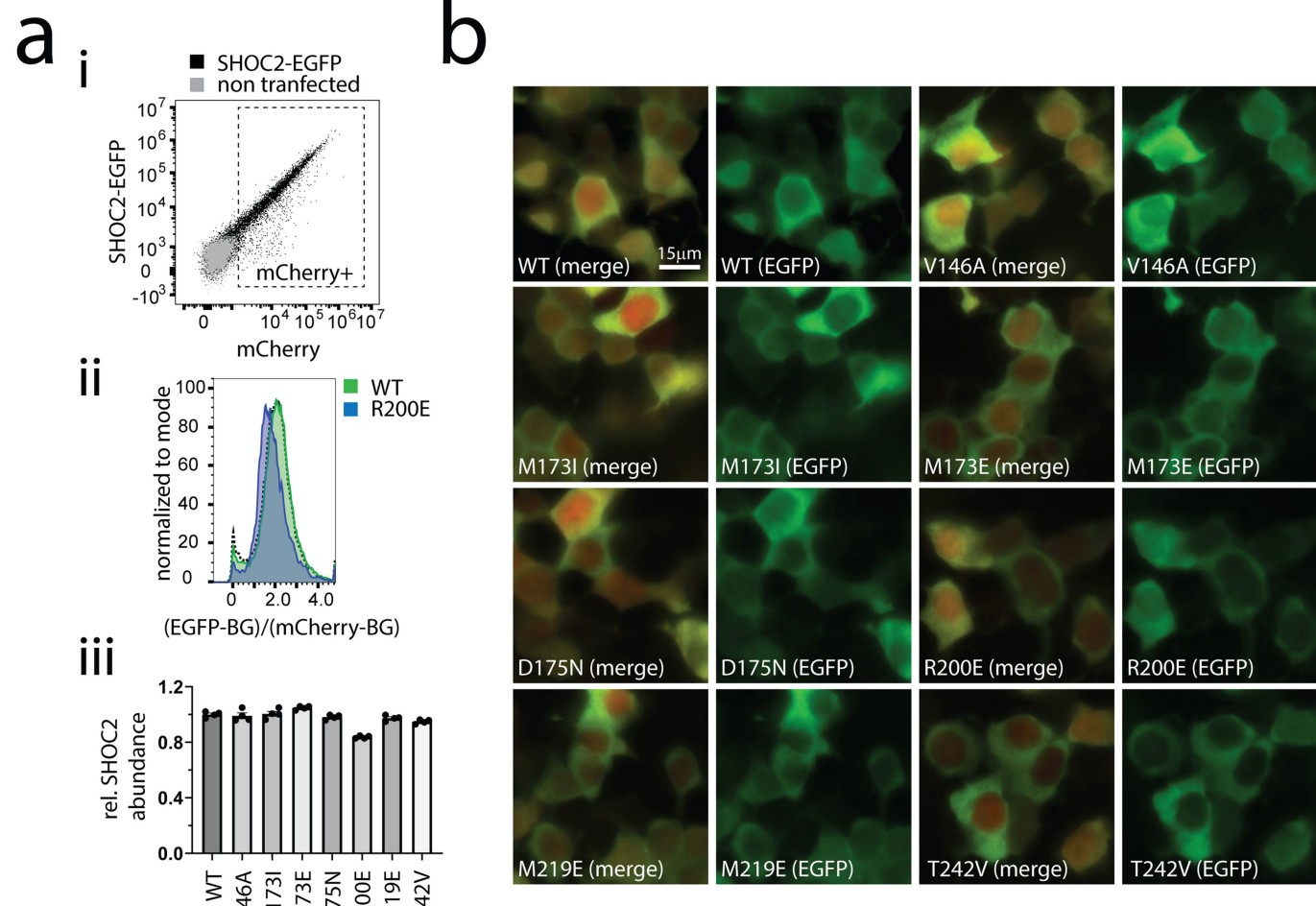

**Extended Data Fig. 6 | SHOC2 mutations do not affect protein stability or localization in cells. a**. Flow cytometry analysis of SHOC2-EGFP-chy-mCherry transfected 293T cells to compare SHOC2-EGFP abundance/stability between the various GOF and LOF variants in cells. (i) Flow cytometry SHOC2-EGFP vs mCherry dot plot, overlay of SHOC WT and non-transfected control, mCherry+ gate is indicated. (ii) mCherry+ cells were selected to extract the ratio of EGFP vs mCherry signal per cell, and for both signals the respective median EGFP or mCherry signal from non-transfected cells (grey population) was subtracted as background (BG) (EGFP-BG)/(mCherry-BG) and plotted as histogram normalized to mode to account for various cell numbers between WT and R200E transfected cells. (iii) Bar graph representing rel. SHOC2-EGFP

abundance of indicated mutants normalized to SHOC2 WT, data represents mean +/− SD of extracted median (EGFP-BG)/(mCherry-BG) values as described in ii from N = 4 independent transfections (individual values per repeat plotted on bar represent median from 1636 < mCherry+ cells < 40098, Avg 20460 cells). **b**. Representative immunofluorescence images of SHOC2-EGFP-chy-mCherry transfected 293T cells with WT and corresponding SHOC2 mutants. Overlay images of mCherry and EGFP channels (left) and EGFP channel only images (right) for SHOC2 variants are shown, size bar corresponds to 15 μm for all images. Results are representative of three independent transfections for each condition.

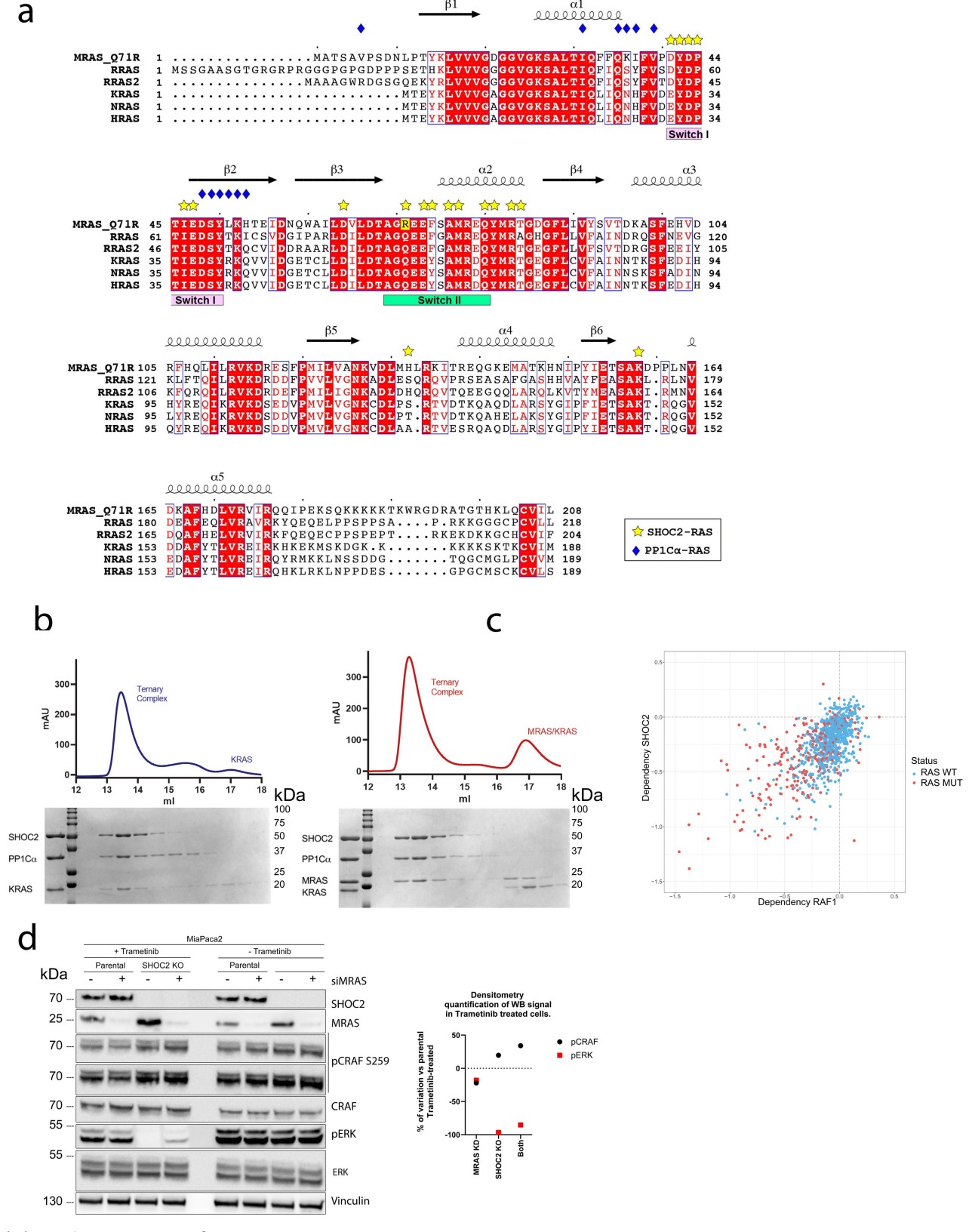

**Extended Data Fig. 7 |** See next page for caption.

**Extended Data Fig. 7 | The RAS-PP1C interface favours MRAS while N/KRAS compensate for MRAS loss. a**. RAS and RAS-related proteins sequence alignment. Amino acid sequences of human RAS-related proteins MRAS Q71R, RRAS, RRAS2 and aligned with KRAS, NRAS and HRAS. Identically conserved residues are shaded red. The site of the Q71R mutation in the present structure is shaded yellow. Secondary structure elements of MRAS are indicated above the alignment. Symbols above the alignment indicate residues that in the holoenzyme complex lie in the interface with SHOC2 (yellow stars) or PP1Cα (blue diamonds). **b**. SHOC2, PP1Cα, and KRAS (GppNHp) were mixed in a 1:1:1.2 molar ratio and subjected to size-exclusion chromatography using a Superdex 200 Increase 10/300 GL column. In the absence of MRAS, the ternary complex formed with KRAS (left). SHOC2, PP1Cα, MRAS and KRAS were combined (right) in a 1:1:1.2:1.2 molar ratio, respectively, before SEC analysis. MRAS and KRAS were both charged with non-hydrolyzable GTP analogue GppNHp prior to combining with SHOC2 and PP1C. The resulting UV280 absorbance trace is shown in the top panel, and Coomassie-stained SDS-PAGE gel of the corresponding fractions is shown in the lower panel. Note that the ternary complex assembles with MRAS, while KRAS and excess MRAS elutes as expected for the unbound GTPases. The data are representative of two independent experiments. **c**. A scatter plot with the dependency score of RAF1 on the x axis and the dependency score of SHOC2 on the y axis. The dashed lines indicate a dependency score of zero (no dependency). A highly negative dependency score implies that a given cell line is highly dependent on that gene. Cell lines dependent on both SHOC2 and RAF1 are indicated in the lower left of the scatter plot. **d**. Western Blot analysis of MiaPaca2 parental cells or SHOC2 KO cells treated with trametinib (10 nM) for 24 h and siRNA targeting MRAS (100 nM) for 48 h. Densitometry quantification of pCRAF(S259)/CRAF and pERK/ERK levels from Western Blot analysis normalized to trametinib treated parental cells. Samples were derived from the same experiment and blots were processed in parallel. Representative images shown are from two independent experiments.

**Extended Data Table 1 | Data collection and refinement statistics (molecular replacement) for structures 7TXH and 7TYG**

|  | Apo SHOC2(80-582) | SHOC2(80-582)-MRAS-PP1Cα |
|---|---|---|
| **Data collection** |  |  |
| Space group | C121 | P1211 |
| Cell dimensions |  |  |
| $a$, $b$, $c$ (Å) | 91.4, 102.9, 120.9; | 73.0, 115.8, 137.8; |
| $\alpha$, $\beta$, $\gamma$ (°) | 90, 101.7, 90 | 90, 100.3, 90 |
| Resolution (Å) | 118.3-1.9 (1.902-1.895) | 88.0-1.95 (1.958-1.951) |
| $R_{sym}$ or $R_{merge}$ | 0.051 (0.539) | 0.077 (0.740) |
| $I / \sigma I$ | 13.0 (2.0) | 10.4 (1.9) |
| Completeness (%) | 94.7 (100.0) | 99.9 (100.0) |
| Redundancy | 3.3 (3.3) | 4.3 (4.4) |
|  |  |  |
| **Refinement** |  |  |
| Resolution (Å) | 39.27-1.9 | 36.34-1.95 |
| No. reflections | 81447 | 163269 |
| $R_{work}$ / $R_{free}$ | 0.1881/0.2254 | 0.1687/0.2103 |
| No. atoms | 8414 | 17349 |
| Protein | 7788 | 15418 |
| Ligand/ion | 1 | 186 |
| Water | 625 | 1715 |
| $B$-factors | 40.44 | 36.29 |
| Protein | 40.27 | 35.60 |
| Ligand/ion | 30.68 | 40.79 |
| Water | 342.54 | 42.0 |
| R.m.s. deviations |  |  |
| Bond lengths (Å) | .007 | .008 |
| Bond angles (°) | .86 | .90 |
| Ramachandran Favored Regions (%) | 95.55 | 93.0 |
| Additional Allowed Regions (%) | 4.40 | 6.1 |
| Ramachandran Outliers (%) | 0.05 | 0.9 |

*Values in parentheses are for highest-resolution shell.

**Extended Data Table 2 | Affinity of CRAF RBD to different RAS constructs by SPR**

|  | NRAS Q61R | MRAS WT | MRAS Q71R |
| --- | --- | --- | --- |
| CRAF RBD SPR KD (nM) | 147/197 | 322/374 | 213/266 |

Affinity of cRAF1-RBD (aa 31-151) to different avi-tagged and GppNHp -loaded MRAS constructs (immobilized on chip) was quantified by SPR to confirm similar affinity as to canonical RAS isoforms, validating the approach to set up respective TR-FRET assays. Individual results for N=2 independent experiments are given.

**Extended Data Table 3 | Competition IC50 values for untagged proteins in different RAS/RBD TR-FRET assays**

| TR-FRET Pair | IC50 obtained for untagged proteins (nM) | | | | | |
| --- | --- | --- | --- | --- | --- | --- |
| | RBD | FL SHOC2 | FL SHOC2 M173I | MRAS WT | MRAS Q71R | NRAS Q61R |
| MRAS WT/RBD | 351 (66) | 1920 (568) | 1897 / 947 | 981 / 800 | 638 / 409 | 462 (75) |
| MRAS Q71R/RBD | 311 (59) | 913 (307) | 550 / 539 | 1275 / 1043 | 731 / 540 | 537 (111) |
| NRAS Q61R/RBD | 260 (60) | 1593 (307) | 512 / 551 | 1162 / 963v | 660 / 533 | 589 (119) |
| n | 3 | 3 | 2 | 2 | 2 | 3 |

Competition IC50 values obtained by titrating untagged proteins in TR-FRET assays set up between different avi-tagged/ biotinylated and GppNHp -loaded RAS constructs (NRAS Q61R, MRAS and MRAS Q71R) and His-tagged cRAF1-RBD (aa 31-151), using anti-His-Eu as donor and streptavidin-Cy5 as acceptor dyes. IC50 values for untagged RBD and RAS constructs are in line with RAS-RBD affinities as quantified by SPR, validating the sensitivity of the assays. Untagged SHOC2 and SHOC2 M173I constructs displace the RAS-RBD interaction with IC50 values in line with their respective affinities to RAS, confirming that RBD and SHOC2 are competing for binding to RAS. For experiments run N=2, both IC50 values are given. For N=3, the mean value is given with S.D. in brackets.

# Reporting Summary

## Statistics

For all statistical analyses, confirm that the following items are present in the figure legend, table legend, main text, or Methods section.

| n/a | Confirmed | |
|-----|-----------|---|
| ☐ | ☒ | The exact sample size (*n*) for each experimental group/condition, given as a discrete number and unit of measurement |
| ☐ | ☒ | A statement on whether measurements were taken from distinct samples or whether the same sample was measured repeatedly |
| ☒ | ☐ | The statistical test(s) used AND whether they are one- or two-sided<br>*Only common tests should be described solely by name; describe more complex techniques in the Methods section.* |
| ☒ | ☐ | A description of all covariates tested |
| ☒ | ☐ | A description of any assumptions or corrections, such as tests of normality and adjustment for multiple comparisons |
| ☐ | ☒ | A full description of the statistical parameters including central tendency (e.g. means) or other basic estimates (e.g. regression coefficient) AND variation (e.g. standard deviation) or associated estimates of uncertainty (e.g. confidence intervals) |
| ☒ | ☐ | For null hypothesis testing, the test statistic (e.g. *F*, *t*, *r*) with confidence intervals, effect sizes, degrees of freedom and *P* value noted<br>*Give P values as exact values whenever suitable.* |
| ☒ | ☐ | For Bayesian analysis, information on the choice of priors and Markov chain Monte Carlo settings |
| ☒ | ☐ | For hierarchical and complex designs, identification of the appropriate level for tests and full reporting of outcomes |
| ☐ | ☒ | Estimates of effect sizes (e.g. Cohen's *d*, Pearson's *r*), indicating how they were calculated |

*Our web collection on statistics for biologists contains articles on many of the points above.*

## Software and code

Policy information about availability of computer code

| Data collection | SEDFIT version 15.01b<br>Rockmaker Version 3.14.6.6, 3.17.8.1<br>FUSION FX7 system integrated software<br>Unicorn 7.1 |
|---|---|
| Data analysis | Gussi<br>Graphpad Prism 8, 9.2.0<br>Biacore Evaluation  3.0.12.15655<br>FloJo V10.7.1<br>XDS<br>autoPROC<br>Phaser 1.18.2_3874+SVN<br>Coot molecular graphics 0.96 EL<br>Phenix 1.19.2_4158<br>Buster 2.11.8<br>Depmap package v1.8<br>Experiment hub package V2.2 (Bioconductor)<br>R 4.1.1<br>Modelling packages (see methods for references):<br>3D-RISM<br>Amber ff19SB |

```
parm@Frosst
AMBER 16
PMEMD CUDA
CPPTRAJ
Getcontacts
```

For manuscripts utilizing custom algorithms or software that are central to the research but not yet described in published literature, software must be made available to editors and reviewers. We strongly encourage code deposition in a community repository (e.g. GitHub). See the Nature Portfolio guidelines for submitting code & software for further information.

## Data

Policy information about availability of data

All manuscripts must include a data availability statement. This statement should provide the following information, where applicable:
- Accession codes, unique identifiers, or web links for publicly available datasets
- A description of any restrictions on data availability
- For clinical datasets or third party data, please ensure that the statement adheres to our policy

The data supporting the findings of this study are available from the corresponding authors upon reasonable request. Structural factors and coordinates have been deposited in the Protein Data Bank under accession codes 7TYG for the SHOC2 (80-582) apo structure and 7TXH for the SHOC2 M173I (80-582)-MRAS Q71R (1-178)-PP1Ca (7-300) ternary complex structure.

## Human research participants

Policy information about studies involving human research participants and Sex and Gender in Research.

| | |
|---|---|
| Reporting on sex and gender | N/A |
| Population characteristics | N/A |
| Recruitment | N/A |
| Ethics oversight | N/A |

Note that full information on the approval of the study protocol must also be provided in the manuscript.

# Field-specific reporting

Please select the one below that is the best fit for your research. If you are not sure, read the appropriate sections before making your selection.

☒ Life sciences ☐ Behavioural & social sciences ☐ Ecological, evolutionary & environmental sciences

For a reference copy of the document with all sections, see nature.com/documents/nr-reporting-summary-flat.pdf

# Life sciences study design

All studies must disclose on these points even when the disclosure is negative.

| | |
|---|---|
| Sample size | Sample sizes for individual experiments were determined with standard practices in the respective techniques. The number of independent experiments for each type of experiment is noted in the manuscript. Crystallographic data was determined from single crystals for each respective structure. |
| Data exclusions | No data was excluded from this manuscript. For images or experiments that are representative of independent replicates, the number of independent experiments has been noted. |
| Replication | We were able to replicate all data present in the manuscript. |
| Randomization | We did not conduct experiments where randomization was applicable. |
| Blinding | We did not conduct experiments where blinding was applicable. |

# Reporting for specific materials, systems and methods

We require information from authors about some types of materials, experimental systems and methods used in many studies. Here, indicate whether each material, system or method listed is relevant to your study. If you are not sure if a list item applies to your research, read the appropriate section before selecting a response.

## Materials & experimental systems

| n/a | Involved in the study |
|---|---|
| ☐ | ☒ Antibodies |
| ☐ | ☒ Eukaryotic cell lines |
| ☒ | ☐ Palaeontology and archaeology |
| ☒ | ☐ Animals and other organisms |
| ☒ | ☐ Clinical data |
| ☒ | ☐ Dual use research of concern |

## Methods

| n/a | Involved in the study |
|---|---|
| ☒ | ☐ ChIP-seq |
| ☐ | ☒ Flow cytometry |
| ☒ | ☐ MRI-based neuroimaging |

## Antibodies

| Antibodies used | LANCE Eu-W1024 Anti-6x his, Perkin Elmer # AD0400 / # AD0401 / # AD0402;<br>Anti-BRAF pS365, Cell Signaling Technologies # 921S;<br>Anti BRAF pS729, Abcam # ab124794;<br>Rabbit anti-MRAS, Abcam # ab176570;<br>Mouse anti-Vinculin, Sigma # V9131;<br>Rabbit anti-SHOC2, Cell Signaling Technology # 53600;<br>Rabbit anti-phospho-p44/42 MAPK (Erk1/2) (Thr202/Tyr204), Cell Signaling Technology # 4370;<br>Rabbit anti-p44/42 MAPK, Cell Signaling Technology # 9102;<br>Rabbit anti-phospho-MEK1/2 (Ser217/221), Cell Signaling Technology # 9154;<br>Rabbit anti-MEK1/2, Cell Signaling Technology # 9122;<br>Rabbit anti-phospho-C-Raf (Ser259), Cell Signaling Technology # 9421;<br>Rabbit anti-C-Raf, Cell Signaling Technology # 53745;<br>Rabbit anti-Flag-tag, Cell Signaling Technology # 14793;<br>Anti-mouse IgG HRP linked, Cell Signaling Technology # 7076;<br>Anti-rabbit IgG HRP linked, Cell Signaling Technology # 7074; |
|---|---|
| Validation | Antibodies were not orthogonally validated in-house. Antibodies with as many trusted citations as possible were used. All antibodies functioned as expected for their respective assays. |

## Eukaryotic cell lines

Policy information about cell lines and Sex and Gender in Research

| Cell line source(s) | HEK293T from ThermoFisher Scientific. MiaPaca2 cells were sourced from ATCC. cells were sourced from SF9 and SF21 cells were from Expression systems. |
|---|---|
| Authentication | Cell lines were not further authenticated. |
| Mycoplasma contamination | HEK293T and MiaPaca2 cells tested negative for mycoplasma contamination. SF9 and SF21 were not tested for contamination. |
| Commonly misidentified lines<br>(See ICLAC register) | No commonly misidentified cell lines were used in this study. |

## Flow Cytometry

### Plots

Confirm that:

☒ The axis labels state the marker and fluorochrome used (e.g. CD4-FITC).

☒ The axis scales are clearly visible. Include numbers along axes only for bottom left plot of group (a 'group' is an analysis of identical markers).

☒ All plots are contour plots with outliers or pseudocolor plots.

☒ A numerical value for number of cells or percentage (with statistics) is provided.

### Methodology

| Sample preparation | 16k HEK293T cells were transfected with 120ng plasmid in 96 well format 1 day post seeding.  48h after transfection cells were prepared from 96 well standard tissue culture plates for flow cytometry. |
|---|---|
| Instrument | CytoFLEX S (Beckman Coulter) |

| Software | FloJo V10.7.1 |
|---|---|
| Cell population abundance | Cell sorting was not used in this study. |
| Gating strategy | Gating was based on populations of singlet cells which expressed mCherry signal (shown in extended date figure 6a panel i). To select these populations, transfected cells were compared to untransfected cells to determine minimal mCherry signal for the gate. Please note this is shown in the extended data rather than in Supplementary Information. |

☒ Tick this box to confirm that a figure exemplifying the gating strategy is provided in the Supplementary Information.

