## [Peer Review File · Nature]

Manuscript Title: Structure of the MRAS-SHOC2-PP1C phosphatase complex

Reviewer Comments & Author Rebuttals

Reviewer Reports on the Initial Version:

Referees' comments:

Referee #1 (Remarks to the Author):

In this manuscript, Hauseman et al., present X-ray structures of the SHOC2-PP1C-MRAS complex and of the SHOC2 protein alone, as well as additional structural, biophysical and biochemical data on the functional regulation of the complex. The SHOC2-PP1c-MRAS complex is well-known to dephosphorylate a critical negative regulatory phosphorylation site in members of the RAF family of proteins (S259 in CRAF) and substantial insight has been gained by previous biochemical work, including the study of recurrent mutations in genes encoding for components of the complex in Noonan-like syndrome. In this study, a high resolution (1.95 Å) X ray crystal structure of the complex is accompanied by biophysical (SPR) studies. These studies revealed that a) PP1C α requires the simultaneous presence of MRAS and SHOC2 to participate in the complex and b) that recurrent mutations in SHOC2 or MRAS found in patients cooperate in promoting formation of the complex.

Overall, even though most of the conclusions have been already inferred by a number of previous biochemical studies (Rodriguez-Viciana et al, Mol Cell, 2006, as well as more recent studies on the topic by Young et al., PNAS 2018, Boned del Rio et al PNAS, 2019), this is a well-conducted study and provides valuable structural information. There are however certain issues that need to be addressed, to enhance the significance of the study.

1. The authors identified LOF and GOF mutations predicted to alter interactions based on the X ray structure and validated their findings in a mammalian two-hybrid assay monitoring the SHOC2-MRAS interaction. Orthogonal validation of the structural findings by functional cell-based experiments examining RAF dephosphorylation, RAF activating phosphorylation and dimerization and MAPK activation upon expression of the same LOF and GOF mutant forms of the components of the complex would strengthen the study.

2. Major question in the field, and a topic of controversy in previous studies, is the relative contribution of different RAS isoforms to RAF dephosphorylation at S259. It is somewhat puzzling that the authors did not detect a preferential binding of MRAS to the complex, over KRAS/NRAS/HRAS. The authors further propose, based on their findings, that two different RAS molecules are needed to carry out RAF activation (one to bind SHOC2-PP1c, the other RAF-14-3-3). Cell-based and biochemical studies to support their proposed model and determine what is the relevant contribution of each RAS isoform in cells would strengthen the study. For example, what are the complexes of MRAS versus KRAS/NRAS/HRAS with SHOC2-PP1C that are formed in living cells and what is the relative contribution of each RAS to RAF dephosphorylation and to RAF recruitment to the membrane and activating phosphorylation? To what extent is RAFS259 dephosphorylated in

cells in the absence of MRAS or other RAS isoforms?

Referee #2 (Remarks to the Author):

The RAS-MAPK pathway controls the proliferation and survival of a diversity of cell types during the development of multicellular organisms. Its dysregulation is closely associated with tumor development and progression. Activation of RAF by the small RAS GTPases is one of the key events in this pathway. The underlying mechanism is complex and still poorly understood at some levels. In particular, it involves dephosphorylation of a critical serine residue in the N-terminal region of RAF kinases (Ser 259 in Raf1) and which is part of a phospho-dependent binding site for 14-3-3 proteins that repress RAF activity. Dephosphorylation of this site by the SHOC2-PP1Ca-MRAS complex releases 14-3-3 binding and thereby contributes to RAF activation. In this study, the authors undertook the structural characterization of the ternary complex comprising these three proteins (SHOC2, PP1Ca and MRAS).

First, using purified proteins to reconstitute the complex in vitro, they showed by SPR that the complex forms cooperatively. They also showed that some RASopathy mutations found in SHOC2 and MRAS enhance binding between the three proteins. The stoichiometric association between the three proteins was also demonstrated by analytical centrifugation and gel filtration. Given their improved ability to form a stable complex, the authors elected to use RASopathy mutant versions of SHOC2 (M173I) and MRAS (Q71R) to structurally characterize the complex.

Using x-ray diffraction, the authors next determined the crystal structure of the complex at a resolution of 1.95Å. They then described its topological organization, identified key interacting residues at each interface, explained how SWI and II movement imparted by GTP-bound MRAS contributes to ternary complex formation, and speculated on the role of water molecules in enabling cooperative complex assembly.

Consistent with the conservation of MRAS residues involved in SHOC2 binding with those of other RAS GTPases, the authors showed by SPR the ability of K and NRAS to interact with SHOC2 in a GTP-dependent manner. This result suggests that other RAS GTPases could also assemble with SHOC2-PP1Ca to promote RAF activation through a mechanism similar to that of MRAS.

Finally, the authors noted that several mutations affecting SHOC2, PP1Ca or MRAS found in RASopathies and which are thought to enhance the activity of the complex, are mainly located at the interfaces between the three members. Inspection of the crystal structure suggests that some of these mutations act by stabilizing interactions between specific residues and thus confer greater stability to the complex. A two-hybrid assay in mammalian cells that tested a limited set of mutations showed consistent results with their hypothesis.

Overall, this is an interesting study that describes the atomic structure of a protein complex that is very relevant in cancer biology. This work now provides a structural understanding as to how GTP binding on MRAS promotes complex formation as well as the mechanism by which certain

Rasopathy mutations act. Ultimately, this structure should facilitate the development of anticancer therapeutic tools targeting the SHOC2-PP1Ca-MRAS complex. Despite its usefulness for RAS researchers, this work remains fairly descriptive and would definitely gain by mechanistically and functionally exploring some of the hypotheses put forward by the authors. The following aspects could be further explored.

1- They propose that most RASopathy mutations work by increasing complex stability. Yet only two such mutations (SHOC2_M173I and MRAS_Q71R) have been properly tested. They should test a few more to solidify their claim. As a matter of fact, 2-hybrid assay in mammalian cells is interesting, but is limited: a- is does not reflect the physiological environment of these proteins, b- proteins are tested as binary combinations, which might not adequately capture the intrinsic binding properties of the ternary complex, c- they do not demonstrate equal protein production and localization, and d- the assay does not provide any information as to whether the mutations genuinely confers an autonomous gain of activity to the complex. A proper in vitro functional assay should be developed to demonstrate the gain of activity towards RAF.

2- Their work suggests that other RAS GTPases also engage in complex formation with SHOC2-PP1Ca in a GTP-dependent manner. This is an exciting possibility and their study would have greater significance in demonstrating that such complexes exist physiologically and promote RAF dephosphorylation (and activation).

3- An interesting hypothesis from their work is that at least two distinct RAS complexes might be required for RAF activation, namely, one that recruits RAF and one that dephosphorylate the N-terminal 14-3-3 binding site on RAF. As above, their work would be enhanced by providing experimental evidence for this.

4- The authors have developed very nice in vitro reconstitution assays to study the assembly of the SHOC2-PP1Ca-MRAS complex. However, apart from the determination of dissociation constants, the authors did not exploit these biochemical tools to investigate the functional consequences of complex formation. For example, the impact of complex formation and especially the specific impact of MRAS binding on PP1Ca activity was not addressed. At least two non-mutually exclusive hypotheses could explain the RAS-dependent regulation of RAF dephosphorylation by PP1Ca: 1) colocalization of RAS-SHOC2-PP1Ca and RAS-RAF complexes at the plasma membrane and/or 2) allosteric activation of the PP1Ca-containing holoenzyme upon RAS binding. It would be interesting for the authors to investigate the functional consequence of RAS-PP1Ca contacts in terms of PP1Ca catalytic output. The design and testing of point mutations at the PP1Ca-MRAS interface would be key for such experiments. For example, the authors could introduce the divergent amino acids found at orthologous positions of other RAS proteins (e.g. RRAS, NRAS, KRAS) at the RAS-PP1Ca interface (p7. Lines 4-6) to test the hypothesis.

Minor points:

1- Fig. 4c: G28V should read G23V

2- On page 6, line 25: (Figure 3b, S3c) should read (Figure 2d, S4c)

3- On page 11, lines 21-23: Figure 2c does not reflect the statement in the sentence.

Referee #3 (Remarks to the Author):

The RAS-RAF-MAPK cascade is vital for growth factor cell signaling and regulates many biological processes. Previous works have shown that the MRAS-SHOC2-PP1 phosphatase complex plays critical roles in RAF activation, and germline gain-of-function mutations of this complex result in congenital RASopathy syndromes. The current manuscript by Hauseman et al. presents a 1.95 Å high-resolution X-ray crystal structure of the ternary MRAS-SHOC2-PP1c complex and reveals major interfaces of this important complex, which should be technically solid. The crystal structure and related biochemical analysis provides key insights into how some of the SHOC2, MRAS mutants may lead to RASopathy syndromes. This work also provide evidence that GTP-bound active form of RAS in particular M-RAS is critical for formation of the MRAS-SHOC2-PP1c complex. However, it remains unclear how PP1c dephosphorylates RAF

Specific issues:

1. Based on the crystal structure, authors proposed that “two independent RAS-GTP molecules must independently recruit RAF-14-3-3 and SHOC2-PP1c ...” (in the abstract). It may be premature to emphasize “independently” without providing direct evidence for this. In fact, a RAS-RAF signalosome model has been recently proposed in which dimerization/oligomerization of KRAS plays a key role in RAS-RAF interaction (Mysore et al. NSMB 2021). The statement in the current abstract is questionable.
2. The N-terminal truncated SHOC2(80-582) is used for crystallographic studies. This should be clarified in the main text and main figure legends. Was SHOC2(80-582) or FL-SHOC2 used in other biochemical (in particular binding) analysis? Please clarify.
3. Would the missing N-terminal region of SHOC2 play a role in forming the MRAS-SHOC2-PP1c complex?
4. In Table 1, at least for crystallographic data collection, highest resolution bin parameters should be shown separately.
5. Given the resolution and crystallographic statistics, the crystal structural model should be reliable. For Ramachandra plot outlier residues (Table 1), to evaluate if their deviations from standard stereochemistry are functionally relevant, their positions in the complex structure and related electron density maps should be provided in Supplementary Info.
6. Table 1, Completeness listed twice.
7. Sup. Fig 1., color spectrum bar indicating conservation level should be included;

Author Rebuttals to Initial Comments:

Hauseman et al. (Nature manuscript 2022-01-01055A)

“Cooperative assembly and structure of the RAS-SHOC2-PP1c holophosphatase provides insights into the RAS signalosome and disease-relevant mutations”

Response to the reviewers

We sincerely thank all three referees for the positive and constructive comments. We have considered and addressed experimentally whenever possible each comment and we believe that these extensive revisions have substantially improved the manuscript. **Please find below our point-by-point answers**, including new data (“Figure for referee” or “new manuscript Figure”) and text changes that can also be found referenced in the revised manuscript.

Referee #1 (Remarks to the Author):

In this manuscript, Hauseman et al., present X-ray structures of the SHOC2-PP1C-MRAS complex and of the SHOC2 protein alone, as well as additional structural, biophysical and biochemical data on the functional regulation of the complex. The SHOC2-PP1c-MRAS complex is well-known to dephosphorylate a critical negative regulatory phosphorylation site in members of the RAF family of proteins (S259 in CRAF) and substantial insight has been gained by previous biochemical work, including the study of recurrent mutations in genes encoding for components of the complex in Noonan-like syndrome. In this study, a high resolution (1.95 Å) X ray crystal structure of the complex is accompanied by biophysical (SPR) studies. These studies revealed that a) PP1Cα requires the simultaneous presence of MRAS and SHOC2 to participate in the complex and b) that recurrent mutations in SHOC2 or MRAS found in patients cooperate in promoting formation of the complex.

Overall, even though most of the conclusions have been already inferred by a number of previous biochemical studies (Rodriguez-Viciano et al, Mol Cell, 2006, as well as more recent studies on the topic by Young et al., PNAS 2018, Boned del Rio et al. PNAS, 2019), this is a well-conducted study and provides valuable structural information. There are however certain issues that need to be addressed, to enhance the significance of the study.

We thank the reviewer for recognizing the value of the new high-resolution structures of the RAS-SHOC2-PP1 holophosphatase provided in our study. We have addressed the reviewer’s points as detailed below and the revisions have overall strengthened our findings.

1. The authors identified LOF and GOF mutations predicted to alter interactions based on the X ray structure and validated their findings in a mammalian two-hybrid assay monitoring the SHOC2-MRAS interaction. Orthogonal validation of the structural findings by functional cell-based experiments examining RAF dephosphorylation, RAF activating phosphorylation and dimerization and MAPK activation upon expression of the same LOF and GOF mutant forms of the components of the complex would strengthen the study.

We agree with the reviewer that direct functional validation of the SHOC2 mutants inferred from the X-ray ternary structure would improve the study. Please note that for all the SHOC2 mutants we already show in cells an altered affinity towards MRAS, which goes in the predicted / expected direction (lower affinity for LoF and higher for GoF mutants, New Fig. 3c), but as correctly pointed out direct evidence for pathway modulation is missing. Referee #2 had a somewhat related request in her/his comment n.1.

To address this question, we have generated the following new data:

Cellular experiments showing that SHOC2 LoF / GoF mutant expression results in predicted **MAPK-pathway modulation**: respectively an increase (LoF) and decrease (WT and GoF) in the

inhibitory S259 phosphorylation on CRAF (RAF inhibition) and a consequent decrease (LoF) and increase (WT and GoF) MEK / ERK activating phospho-sites (new Fig. 5d).

New Fig.5d: Immunoblot analysis of MiaPaca2 parental cells (Par), SHOC2 KO (KO) and stable cell lines reconstituted with SHOC2 mutants, after 10nM Trametinib treatment for 1h or 24 h (+). Densitometry quantification (% variation) of p-CRAF/CRAF and p-ERK/ERK levels from Immunoblot analysis normalized to untreated MiaPaca2 parental cells.

- Cellular experiments showing that the **protein abundance/stability** of the SHOC2 LoF / GoF mutants is comparable to that of SHOC2 wild-type protein and that we do not observe any aggregation in cells. (new Extended Data Figure 6).

New Extended Data Figure 6. (a) Flow cytometry analysis of SHOC2-EGFP-chy-mCherry transfected 293T cells to compare SHOC2-EGFP stability between the various GOF and LOF variants in cells. (i) SHOC2-EGFP vs mCherry dot plot overlay of SHOC WT and non-transfected control, (ii) mCherry+ cells were selected to extract the ratio of EGFP vs mCherry signal and for both signals the median EGFP or mCherry signal from non-transfected cells was subtracted $(EGFP-BG)/(mCherry-BG)$, respectively (BG) and plotted as histogram normalized to mode to account for various cell numbers between WT and R200E. (iii) Bar graph representing rel. SHOC2-EGFP abundance of indicated mutants normalized to SHOC2 WT, data represents mean \pm SD of extracted median $(EGFP-BG)/(mCherry-BG)$ values as described in ii from $N = 4$ independent repeats. (b) Representative immunofluorescence images of SHOC2-EGFP-chy-mCherry transfected 293T cells with WT and corresponding SHOC2 mutants. Overlay images of mCherry and EGFP channels (left) and EGFP channel only images (right) for SHOC2 variants are shown, size bar corresponds to 15 μ m for all images.

mutants normalized to SHOC2 WT, data represents mean \pm SD of extracted median $(EGFP-BG)/(mCherry-BG)$ values as described in ii from $N = 4$ independent repeats. (b) Representative immunofluorescence images of SHOC2-EGFP-chy-mCherry transfected 293T cells with WT and corresponding SHOC2 mutants. Overlay images of mCherry and EGFP channels (left) and EGFP channel only images (right) for SHOC2 variants are shown, size bar corresponds to 15 μ m for all images.

- Additionally, for the disease relevant SHOC2 M173I GoF mutant (found in Rasopathies) we have produced: (a) MD simulations that model the structural changes caused by the M173I mutation and that help interpret the observed gain in affinity towards RAS (new Fig.1a, Extended Data Figure 5); (b) in vitro activity validation by showing specific enhanced site-dependent de-phosphorylation of BRAF S365, but not of S729 site on recombinant BRAF inactive complex (new Fig.5b)

New Extended Data Fig. 5: Representative structures of the SHOC2-MRAS interface from simulated SHOC2_WT-MRAS_WT-PP1Ca ternary complex. Three distinct states of the MRAS (dark blue) - SHOC2 (purple) interface are observed from the clustered MRAS_WT, SHOC2_WT, PP1Ca_WT trajectories and aligned to the crystal structure of the ternary SHOC2_M173I (pale green), MRAS_Q71R (grey), PP1Ca_WT (not

shown) complex. (a) In state 1, SHOC2 M173 pushes the sidechain of MRAS M77 away from the interface, while leaving SHOC2 Y198 unperturbed. (b) In state 2, M173 is buried, resulting in the displacement of Y198, altering the hydrogen bonding network compared to that of the crystal structure. (c) In state 3, M173 directly pushes the MRAS switch II loop away from the SHOC2 interface.

New Fig. 5b. In vitro dephosphorylation of autoinhibited and active-state BRAF/MEK1/14-3-3 complexes by the SHOC2 holophosphatase. Purified full-length BRAF complexes in either the autoinhibited state (left panel) or active, dimeric state (right panel) were incubated with the indicated ternary SHOC2 complexes, with PP1Ca alone, or with lambda phosphatase (control) and western-blotted with phospho-specific antibodies for pS365 and pS729. A Coomassie-stained gel is shown to confirm equivalent loading of the BRAF complexes. Note that the pS365 site (equivalent to pS259 in CRAF) is selectively dephosphorylated relative to the pS729 site (equivalent to pS621 in CRAF) in the active dimer, while both are relatively protected in the autoinhibited state, where they are both bound by the 14-3-3 dimer.

Overall, the new data validate the structural findings presented in the manuscript and corroborate the concept that changes in affinities between SHOC2-RAS result in variation in RAS/MAPK pathway activity and ultimately provide the basis to understand disease-relevant mutations, such as those found in Rasopathies.

2. Major question in the field, and a topic of controversy in previous studies, is the relative contribution of different RAS isoforms to RAF dephosphorylation at S259. It is somewhat puzzling that the authors did not detect a preferential binding of MRAS to the complex, over KRAS/NRAS/HRAS.

The authors further propose, based on their findings, that two different RAS molecules are needed to carry out RAF activation (one to bind SHOC2-PP1c, the other RAF-14-3-3). Cell-based and biochemical studies to support their proposed model and determine what is the relevant contribution of each RAS isoform in cells would strengthen the study. For example, what are the complexes of MRAS versus KRAS/NRAS/HRAS with SHOC2-PP1C that are formed in living cells and what is the relative contribution of each RAS to RAF dephosphorylation and to RAF recruitment to the membrane and activating phosphorylation? To what extent is RAFS259 dephosphorylated in cells in the absence of MRAS or other RAS isoforms?

We fully agree with the reviewer that the scattered evidence from previous studies showing SHOC2 interacting preferentially with MRAS, but also with the canonical RAS proteins is clouding the understanding of the exact contribution for each of those isoforms in the SHOC2-PP1 holophosphatase complex. However, the exact relative contribution of each of the RAS isoforms in cells to cite the referee's own words is a "Major question in the field" and to get a definitive answer will take a full panel of experimental approaches and cell models that, in our view, would represent an entire new study *per se*. For example, to interrogate pathway activity one could generate isogenic cell models (e.g. with CRISPR-KO technology) expressing exclusively one of the 4 RAS proteins in question (and to be fully comprehensive also the other 2 RAS-related proteins, RRAS and RRAS2). Furthermore, the relative quantitation of endogenous SHOC2 complexes with canonical & non-canonical RAS proteins in cell models (ideally equally expressing all RAS proteins) with current immuno-precipitation protocols is a nontrivial task, as confirmed by the paucity of published data (mostly in conditions of exogenous expression) and likely due to technical issues, such as the fast intrinsic RAS-GTP hydrolysis acting immediately during the process of cell lysis, washing steps, etc. Hence, our choice to utilize in vitro techniques that take advantage of a non-hydrolysable analogue of GTP for our studies. The above-mentioned cell investigations, although very important, will take a significant amount of time, might not be conclusive / generalizable (different cell models should be used) and we respectfully consider them out of the scope of the current manuscript focused on the structural characterization of the MRAS-SHOC2-PP1 complex.

Nevertheless, we provide here what in our view is undeniable evidence that SHOC2 forms phosphatase complexes with MRAS but also with canonical RAS proteins. Published genetic data and our findings suggests that both types of complexes are relevant for cell physiology by mediating RAF activation to promote MAPK pathway signaling. In detail, the following new data shows:

- (1) Complex Formation: SHOC2 is preferentially forming a complex with MRAS over canonical RAS proteins, due to higher affinity only with respect to a ternary complex with PP1 (as shown in new Figures 1a and 5a, SHOC2-RAS binary complexes alone appear similar). We speculate that this might be due to the portion of MRAS that is in direct contact with PP1 and which is evolutionary not conserved in the canonical RAS proteins (see alignment in Extended Data Fig. 7a).

New Fig. 1a. Cooperative assembly of the SHOC2-MRAS-PP1Ca ternary complex. Affinity (K_D) and associated cooperativity (α) derived from SPR sensorgrams with surface immobilized MRAS-GppNHp (WT or Q71R) and indicated analytes. Cooperativity (α) defined as the ratio of binary and ternary K_D 's, $SHOC2 = K_D^{SHOC2} / K_D^{SHOC2-PP1}$. RU increase consistent with the saturation of MRAS with both proteins R_{MAX} , $SHOC2 \sim 800-1000$ RU; R_{MAX} , $SHOC2/PP1Ca \sim 1250-1500$ RU.

New Fig. 5a. Affinity (K_D) and associated cooperativity (α) derived from SPR sensorgrams with surface immobilized NRAS-GppNHp, NRAS Q61R-GTP, KRAS-GppNHp, or KRAS Q61R-GTP and indicated analytes. Cooperativity (α) defined as the ratio of binary and ternary K_D 's, $SHOC2 = K_D^{SHOC2} / K_D^{SHOC2-PP1}$.

(2) Functional Relevance of SHOC2-PP1 complex formed with the canonical RAS proteins and MRAS relative contribution.

i. We have set up and performed an *in vitro* RAF de-phosphorylation assay which shows that a SHOC2-PP1 complex formed with KRAS is capable, albeit with reduced efficiency than MRAS, to produce site-dependent de-phosphorylation of the BRAF inhibitory S365 site (S259 of CRAF) on inactive recombinant BRAF complex (new Fig.5b, also shown before to address this referee's point.1 on SHOC2 mutants). Additionally, a group from the NCI RAS Initiative has shown that M/H/K/NRAS-SHOC2-PP1C purified complexes are all proficient in de-phosphorylating CRAF and BRAF *in vitro*^[1]. See new Figure 5b, provided above to reviewer 1.

ii. Analysis of large dataset on genetic perturbations in cancer models (>1000 cell lines) available from DepMap (<https://depmap.org/portal/>) clearly show that MRAS depletion does not influence fitness in models where SHOC2 and RAF are essential, while SHOC2 is necessary in context of GTP-activating mutations on canonical RAS proteins (i.e. underlying co-dependency, see new Fig. 5c). These data also illustrate SHOC2-RAF robust co-dependency across cell models (see Extended Data Fig. 7c), highlighting their fundamental functional interaction, which appears independent on a given RAS partner.

C

New Fig. 5c: (Left panel) A scatter plot showing the relationship between the knockdown of SHOC2 and the 4 RAS genes: MRAS, HRAS, NRAS and KRAS. On the x-axis is the dependency score of each RAS gene in the corresponding trellis, and on the y-axis the dependency score of the SHOC2 gene. The dashed lines indicate a dependency score of zero. A highly negative dependency score implies that a given cell line is highly dependent on that gene. Cell lines dependent on both SHOC2 and RAS are indicated in the lower left of the scatter plot. (Right panel) A bar plot with the calculated Pearson correlation coefficient on the y-axis applied to each mutation group (group of cell lines harboring the associated mutation). A higher positive value indicates a stronger positive relationship: the dependency score of SHOC2 decreases/increases in the same lines as the dependency scores of the RAS genes. N is the number of cell lines in each mutation group.

New Extended Data Fig. 7c: A scatter plot with the dependency score of RAF1 on the x axis and the dependency score of SHOC2 on the y axis. The dashed lines indicate a dependency score of zero (no dependency). A highly negative dependency score implies that a given cell line is highly dependent on that gene. Cell lines dependent on both SHOC2 and RAF1 are indicated in the lower left of the scatter plot.

- iii. Cellular experiments suggesting that MRAS is not accountable for SHOC2 effects on the MAPK signaling pathway in contexts where canonical RAS proteins are pushed into a GTP-bound active state through oncogenic point mutations. To show this, we took advantage of a published model for studying SHOC2 effect on the activation of RAF-MEK-ERK, which is MiaPaca2 KRAS G12C

mutant cell model treated with a MEK inhibitor (Trametinib), which at 24h causes a MAPK-pathway rebound due to loss of negative feedback inhibition and increased RAS-GTP loading^[2]. Here we knocked-down MRAS expression and show that this has minimal / no impact on trametinib-driven MAPK-pathway rebound (24h time point), compared to SHOC2 depletion which leads to increased CRAF S259 phosphorylation and consequent suppression of pERK. From this, we conclude that other RAS proteins (most likely KRAS mutant, in this particular case) must be contributing to SHOC2-PP1 function.

New Extended Data Fig.7d: (left panel) Western Blot analysis of MiaPaca2 parental cells or SHOC2 KO cells treated with trametinib (10nM) for 24 h and siRNA targeting MRAS (100nM) for 48 h. (right panel) Densitometry quantification of p-CRAF/CRAF and p-ERK/ERK levels from Western Blot analysis normalized to trametinib treated parental cells.

Similarly, Franck McCormick and collaborators have recently shown that in the context of paradoxical MAPK pathway activation by RAF-inhibitors, MRAS is redundant for pathway activity with other canonical RAS proteins and it becomes essential only upon depletion of all three canonical RAS proteins (RASless MEFs model)^[3].

- (3) Data supporting the “Two RAS theory”. We would like to clarify our message. Our present data indicate that two different RAS molecules are needed to carry out RAF activation, one that recruits RAF to the cell membrane through a RAS-effector region + RAF-RBD interaction and one that recruits through the very same RAS-effector region SHOC2-PP1. While we do not show this parallel recruitment directly in the manuscript, it appears evident from the structural data (ours on RAS-SHOC2-PP1c and published on RAS-RAF) that there is no space for CRAF interaction with RAS within the RAS-SHOC2-PP1 complex. To directly demonstrate this, we performed a competition experiment with a newly established TR-FRET PPI assay between MRAS Q71R and RAF-RBD (of note CRAF-RBD has similar affinity for MRAS Q71R than for NRAS Q61R). SHOC2 titrations fully displace the MRAS Q71R + RAF-RBD interaction (New Fig.4b, c, d).

b

	NRAS Q61R	MRAS WT	MRAS Q71R
CRAF RBD SPR KD (nM)	169	348	240

c

d

TR-FRET Pair	IC50 obtained for untagged proteins (nM)					
	RBD	FL SHOC2	FL SHOC2 M173I	MRAS WT	MRAS Q71R	NRAS Q61R
MRAS WT/RBD	351	1920	1377	890	523	462
MRAS Q71R/RBD	311	913	545	1159	635	537
NRAS Q61R/RBD	260	1593	531	1063	596	589

New Fig.4. (b) Affinity of cRAF1-RBD (aa 31-151) to different avi-tagged and GppNHp -loaded MRAS constructs was quantified by SPR to confirm similar affinity as to canonical RAS isoforms. Values given are the average of n=2 experiments. (c) Untagged FL SHOC2 disrupts the interaction of MRAS Q71R:GppNHp and cRAF1-RBD as assessed in a TR-FRET assay, indicating that Shoc2 and RBD binding to RAS are incompatible. (d) Competition IC50 values obtained by titrating untagged proteins in TR-FRET assays set up between different avi-tagged/ biotinylated and GppNHp -loaded RAS constructs (NRAS Q61R, MRAS and MRAS Q71R) and His-tagged cRAF1-RBD (aa 31-151), using anti-His-Eu as donor and streptavidin-Cy5 as acceptor dyes. IC50 values for untagged RBD and RAS constructs are in line with RAS-RBD affinities as quantified by SPR, validating the sensitivity of the assays. Untagged Shoc2 and SHOC2 M173I constructs displace the RAS-RBD interaction with IC50 values in line with their respective affinities to RAS, confirming that RBD and SHOC2 are competing for binding to RAS.

IC50 data shown are mean of n=3 for RBD, FL Shoc2 and NRAS Q61R, n=2 for FL Shoc2 M173I, MRAS wt and MRAS Q71R

Nevertheless, we do agree with the referee and recognize that, in absence of direct experimental evidence showing the two separate RAS complexes (one interacting with SHOC2-PP1 and one with RAF) functionally interacting with each other in physiological settings, no final claims can be made on the and this remains a “highly probable” model. We have carefully reviewed the text in the manuscript to avoid overstatements.

In conclusion, our experimental revisions as well as analysis of publicly available data from large genetic screens suggest that in normal conditions MRAS is the preferential partner for SHOC2-PP1 complex formation (higher affinity within the ternary complex), while canonical RAS proteins represent the “second choice”, providing potential functional redundancy to the MAPK signaling pathway. They might however become the predominant RAS component of the phosphatase complex over MRAS, upon events that favor their persistence in a GTP-bound active state, such as the presence of oncogenic mutations (particularly on position Q61 and G13). Given that we believe that the exact relative contribution of each RAS protein in the SHOC2-PP1 complex in cells is a topic for a future study, our claims in the manuscript are restricted to stating that SHOC2 “can form functionally active complexes also with canonical RAS isoforms, as illustrated in the case of RAS-mutated cancers”.

Referee #2 (Remarks to the Author):

The RAS-MAPK pathway controls the proliferation and survival of a diversity of cell types during the development of multicellular organisms. Its dysregulation is closely associated with tumor development and progression. Activation of RAF by the small RAS GTPases is one of the key events in this pathway. The underlying mechanism is complex and still poorly understood at some levels. In particular, it involves dephosphorylation of a critical serine residue in the N-terminal region of RAF kinases (Ser 259 in Raf1) and which is part of a phospho-dependent binding site for 14-3-3 proteins that repress RAF activity. Dephosphorylation of this site by the

SHOC2-PP1Ca-MRAS complex releases 14-3-3 binding and thereby contributes to RAF activation. In this study, the authors undertook the structural characterization of the ternary complex comprising these three proteins (SHOC2, PP1Ca and MRAS). First, using purified proteins to reconstitute the complex in vitro, they showed by SPR that the complex forms cooperatively. They also showed that some RASopathy mutations found in SHOC2 and MRAS enhance binding between the three proteins. The stoichiometric association between the three proteins was also demonstrated by analytical centrifugation and gel filtration. Given their improved ability to form a stable complex, the authors elected to use RASopathy mutant versions of SHOC2 (M173I) and MRAS (Q71R) to structurally characterize the complex. Using x-ray diffraction, the authors next determined the crystal structure of the complex at a resolution of 1.95Å. They then described its topological organization, identified key interacting residues at each interface, explained how SWI and II movement imparted by GTP-bound MRAS contributes to ternary complex formation, and speculated on the role of water molecules in enabling cooperative complex assembly. Consistent with the conservation of MRAS residues involved in SHOC2 binding with those of other RAS GTPases, the authors showed by SPR the ability of K and NRAS to interact with SHOC2 in a GTP-dependent manner. This result suggests that other RAS GTPases could also assemble with SHOC2-PP1Ca to promote RAF activation through a mechanism similar to that of MRAS. Finally, the authors noted that several mutations affecting SHOC2, PP1Ca or MRAS found in RASopathies and which are thought to enhance the activity of the complex, are mainly located at the interfaces between the three members. Inspection of the crystal structure suggests that some of these mutations act by stabilizing interactions between specific residues and thus confer greater stability to the complex. A two-hybrid assay in mammalian cells that tested a limited set of mutations showed consistent results with their hypothesis.

Overall, this is an interesting study that describes the atomic structure of a protein complex that is very relevant in cancer biology. This work now provides a structural understanding as to how GTP binding on MRAS promotes complex formation as well as the mechanism by which certain Rasopathy mutations act. Ultimately, this structure should facilitate the development of anticancer therapeutic tools targeting the SHOC2-PP1Ca-MRAS complex. Despite its usefulness for RAS researchers, this work remains fairly descriptive and would definitely gain by mechanistically and functionally exploring some of the hypotheses put forward by the authors. The following aspects could be further explored.

This is an excellent, very comprehensive summary. We thank the reviewer for pointing out the direct implications that our findings can have for developing novel therapeutics tackling large unmet medical needs, like RAS-driven cancers and RASopathies. To address the referee's requests, we are now providing extensive revisions that strengthen the mechanistic validation of our structural findings.

1- They propose that most RASopathy mutations work by increasing complex stability. Yet only two such mutations (SHOC2_M173I and MRAS_Q71R) have been properly tested. They should test a few more to solidify their claim. As a matter of fact, 2-hybrid assay in mammalian cells is interesting, but is limited: a- is does not reflect the physiological environment of these proteins, b- proteins are tested as binary combinations, which might not adequately capture the intrinsic binding properties of the ternary complex, c- they do not demonstrate equal protein production and localization, and d- the assay does not provide any information as to whether the mutations genuinely confers an autonomous gain of activity to the complex. A proper in vitro functional assay should be developed to demonstrate the gain of activity towards RAF.

The referee mentions the limitations of the mammalian-2-hybrid cellular assay. This comment is well taken and admittedly, this is a purely binary binding assay between two proteins in cells (although we cannot exclude that endogenously expressed proteins, like PP1 might play a role in facilitating the ternary complex formation). We note though that this technique was able to faithfully reproduce previously published increased and decreased RAS-binding affinities of respective SHOC2 mutants M173I and D175N^[4]. To

demonstrate equal protein levels we used an orthogonal SHOC2-EGFP bipartite system which allowed for quantitative assessment of SHOC2-EGFP abundance compared to an mCherry expression control. Using flow cytometry, we show that mutations did not alter SHOC2 abundance or result in aggregation in cells (precise distribution in cellular compartments was not analyzed). Please see **New Extended Data Fig. 6, previously shown in this rebuttal to address Referee #1, point #1.**

However, to overcome the above-mentioned limitations of the M2H assay and directly demonstrate the functionality of the mutants in a physiologically relevant context (cells) we have performed additional cellular experiments. Of note, Referee #1 in comment n.1 has also requested this data. We took advantage of a published model for studying SHOC2 effect on the RAF-MEK-ERK axis, which are MiaPaca2 KRAS G12C mutant cells treated with a MEK inhibitor (Trametinib) that at 24h time point causes a MAPK-pathway rebound due to loss of negative feedback loop inhibition and increased RAS-GTP loading. Under these conditions, depletion of SHOC2 protein has been shown to suppress MAPK pathway re-activation^[2]. Thus, we have generated MiaPaca2 SHOC2 KO line, which has been then reconstituted with either SHOC2 WT or any of the other GoF and LoF mutants. In both untreated and, more pronouncedly, in 24h Trametinib treated conditions SHOC2 LoF and GoF expression resulted in predicted / expected decreased or increased MAPK-pathway activity (please see **New Fig. 5d, previously shown in this rebuttal to address Referee #1, point #1).**

Additionally, we have directly addressed the referee's specific request for an in vitro functional assay to validate GoF activity towards RAF. Despite the challenges to reproduce the necessary components of the RAS-RAF signalosome *in vitro*, our collaborator and co-author Prof. Mike Eck has developed in his lab an in vitro functional assay where recombinant BRAF in complex with 14-3-3 can be co-incubated with phosphatase or phosphatase-containing complexes to assess RAF de-phosphorylation at different sites. In this setting, we show that the presence of SHOC2 M173I in the MRAS-PP1 complex enhanced, compared to SHOC2 WT, site-dependent de-phosphorylation of the S365 site (equivalent of S259 on CRAF), but not S729 (S621 on CRAF) on inactive recombinant BRAF complex, but not active BRAF complex (please see **New Fig. 5b, previously shown in this rebuttal to address Referee #1, point #1).**

Additionally, we also performed MD simulations that model the structural changes caused by the M173I mutation and that help interpret the observed gain in affinity towards RAS (**please see New Extended Data Fig.5, previously shown in this rebuttal to address Referee #1, point #1).**

Overall, we believe that our revisions solidify the concept that changes in affinities between SHOC2 and RAS result in variations in RAS/MAPK pathway activity and ultimately provide the basis to understand disease-relevant mutations, such as those found in Rasopathies.

2- Their work suggests that other RAS GTPases also engage in complex formation with SHOC2-PP1Ca in a GTP-dependent manner. This is an exciting possibility and their study would have greater significance in demonstrating that such complexes exist physiologically and promote RAF dephosphorylation (and activation).

We agree with the referee that the ability of SHOC2-PP1Ca to interact in a GTP-dependent manner also with the canonical RAS proteins is indeed a finding of great significance for the field. Of note, further supporting evidence in physiological settings was also requested by Referee #1 in comment #2 (**please see our answers there for a more comprehensive analysis**, particularly on why it is a major challenge to establish experimentally the exact physiological contribution of each SHOC2-PP1 complex formed with the canonical RAS proteins). In brief, we present here what in our view is undeniable evidence that SHOC2 forms phosphatase complexes with MRAS but can also with canonical RAS proteins. Published genetic data and our findings suggests that both types of complexes are relevant for cell physiology by mediating RAF activation to promote MAPK pathway signaling. In detail, our revisions are:

- I. Analysis of large dataset on genetic perturbations in cancer models (>1000 cell lines) available from DepMap (<https://depmap.org/portal/>) clearly show that M-RAS depletion does not influence fitness in models where SHOC2 and RAF are essential, while SHOC2 is necessary in context of GTP-activating mutations on canonical RAS proteins (i.e. underlying co-dependency). These data also illustrate SHOC2-RAF robust co-dependency across cell models, highlighting their fundamental functional interaction, which appears independent on a given RAS partner (**please see New Fig.5c and New Extended Data Fig.7c, previously shown in this rebuttal to address Referee #1, point #2**).
- II. Cellular experiments suggesting that MRAS is not accountable for SHOC2 effects on the MAPK signaling pathway in contexts where canonical RAS proteins are pushed into a GTP-bound active state through oncogenic point mutations (**please see New Extended Data Fig.7d, previously shown in this rebuttal to address Referee #1, point #2**). To show this, we took advantage of a published model for studying SHOC2 effect on the activation of RAF-MEK-ERK, which is MiaPaca2 KRAS G12C mutant cell model treated with a MEK inhibitor (Trametinib), which at 24h causes a MAPK-pathway rebound due to loss of negative feedback inhibition and increased RAS-GTP loading^[2]. Here we knocked-down MRAS expression and show that this has minimal / no impact on trametinib-driven MAPK-pathway rebound (24h time point), compared to SHOC2 depletion which leads to increased CRAF S259 phosphorylation and consequent suppression of pERK. From this, we conclude that other RAS proteins (most likely KRAS mutant, in this particular case) must be contributing to SHOC2-PP1 function. Similarly, Franck McCormick and collaborators have recently shown that in the context of paradoxical MAPK pathway activation by RAF-inhibitors, M-RAS is redundant for pathway activity with other canonical RAS proteins and it becomes essential only upon depletion of all three canonical RAS proteins (RASless MEFs model)^[3].

In conclusion, our experimental revisions as well as analysis of publicly available data from large genetic screens suggest that in normal conditions MRAS is the preferential partner for SHOC2-PP1 complex formation (higher affinity within the ternary complex), while canonical RAS proteins represent the “second choice”, providing potential functional redundancy to the MAPK signaling pathway. They might however become the predominant RAS component of the phosphatase complex over MRAS, upon events that favor their persistence in a GTP-bound active state, such as the presence of oncogenic mutations (particularly on position Q61 and G13). Admittedly, we have used cancer genetic datasets to prove the relevance of SHOC2-PP1 complexes with canonical RAS proteins, which is not exactly a normal physiological setting. We believe that establishing the exact relative contribution of each RAS protein within the SHOC2-PP1 complex in normal physiological settings it is a request that would require major experimental efforts and time and we respectfully suggest that this should be a topic for future studies. Our claims in the manuscript are restricted to state that SHOC2 “can form functionally active complexes also with canonical RAS isoforms, as illustrated in the case of RAS-mutated cancers”.

3- An interesting hypothesis from their work is that at least two distinct RAS complexes might be required for RAF activation, namely, one that recruits RAF and one that dephosphorylate the N-terminal 14-3-3 binding site on RAF. As above, their work would be enhanced by providing experimental evidence for this.

Thanks to the referee for noting this important implication that our study brings to the RAS field. This was recognized also by the other two referees.

We would like to clarify our message. Our present data indicate that two different RAS molecules are needed to carry out RAF activation, one that recruits RAF to the cell membrane through a RAS-effector region + RAF-RBD interaction and one that recruits through the very same RAS-effector region SHOC2-PP1. While we do not show this parallel recruitment directly in the manuscript, it appears evident from the structural data (ours on RAS-SHOC2-PP1c and published on RAS-RAF) that there is no space for CRAF interaction with RAS within the RAS-SHOC2-PP1 complex. To directly demonstrate this, we performed a

competition experiment with a newly established TR-FRET PPI assay between MRAS Q71R and RAF-RBD (of note CRAF-RBD has similar affinity for MRAS Q71R than for NRAS Q61R). SHOC2 titrations fully displace the MRAS Q71R + RAF-RBD interaction (**please see New Fig.4b, c, d, previously shown in this rebuttal to address Referee #1, point #2**). Nevertheless, we do agree with the referee and recognize that, in absence of direct experimental evidence, no final claims can be made and this remains a “highly probable” model. We have carefully reviewed the text in the manuscript to avoid overstatements.

4- The authors have developed very nice in vitro reconstitution assays to study the assembly of the SHOC2-PP1Ca-MRAS complex. However, apart from the determination of dissociation constants, the authors did not exploit these biochemical tools to investigate the functional consequences of complex formation. For example, the impact of complex formation and especially the specific impact of MRAS binding on PP1Ca activity was not addressed. At least two non-mutually exclusive hypotheses could explain the RAS-dependent regulation of RAF dephosphorylation by PP1Ca: 1) colocalization of RAS-SHOC2-PP1Ca and RAS-RAF complexes at the plasma membrane and/or 2) allosteric activation of the PP1Ca-containing holoenzyme upon RAS binding. It would be interesting for the authors to investigate the functional consequence of RAS-PP1Ca contacts in terms of PP1Ca catalytic output. The design and testing of point mutations at the PP1Ca-MRAS interface would be key for such experiments. For example, the authors could introduce the divergent amino acids found at orthologous positions of other RAS proteins (e.g. RRAS, NRAS, KRAS) at the RAS-PP1Ca interface (p7. Lines 4-6) to test the hypothesis.

We thank the reviewer for the well-taken comment. Prompted by the request to expand our *in vitro* analysis, we are now providing the following new experimental data:

- (1) Assessment of cooperativity for ternary complex assembly: New SPR data using MRAS, NRAS, and KRAS, now including Q71R/Q61R mutations to evaluate the SHOC2-RAS binary association as well as the total association of the ternary complex of RAS, SHOC2, and PP1Ca. From this experiment we obtained binding constants and calculated **cooperativity values** for both RAS-SHOC2 (binary complex) and RAS-SHOC2-PP1Ca (ternary complex), in all the above-mentioned conditions. The findings are presented in **New Fig. 1a** (for MRAS) and **New Fig. 5a** for canonical N and KRAS (**previously shown in this rebuttal to address Referee #1, point #2**).
- (2) As suggested by the referee we have investigated mutations at the RAS-PP1 interface and characterized the recurrent PP1 mutation found in Rasopathies P50R (PP1 α equivalent of P49R in PP1 β) in our *in vitro* ternary complex assembly assay presented before (please refer to **new Fig.1a already mentioned in point above**). We find that PP1 P50R mutant has increased affinity for MRAS proteins and significantly increases cooperativity of ternary complex formation. Admittedly, we have not gone further than this, for instance trying to swap “*divergent amino acids found at orthologous positions of other RAS proteins (e.g. RRAS, NRAS, KRAS) at the RAS-PP1Ca interface*”. Thus, the hypothesis that MRAS higher affinity for the SHOC2-PP1 complex is provided by the evolutionary divergent residues interacting with PP1, although highly likely, remains not formally demonstrated and we have carefully reviewed the manuscript to make sure that no overstatements are made.
- (3) Investigation over the “impact of complex formation for MRAS binding on PP1Ca activity”. As the referee correctly pointed out, two hypotheses exist: “1. *Co-localization of RAS-SHOC2-PP1Ca and RAS-RAF complexes at the plasma membrane and/or 2. Allosteric activation of the PP1Ca-containing holoenzyme upon RAS binding*”. Our new *in vitro* de-phosphorylation assay (and unpublished small molecule dephosphorylation assay, shown below) validates the enzymatic activity of the PP1C catalytic domain as well as the assembled complex and found that this is similar in apo and complex-bound formats for nonspecific substrates, hence the allosteric activation hypothesis is unlikely (please see **New Fig. 5b, previously shown in this rebuttal to address Referee #1, point #1**).

Unpublished data. Both the catalytic subunit PP1C and the assembled complex have dose-dependent activity in a non-specific dephosphorylation assay, showing that complex assembly does not materially alter the catalytic activity of the enzyme.

Minor points: Thanks for spotting these errors. They have been fixed in the revised manuscript.

1- Fig. 4c: G28V should read G23V

2- On page 6, line 25: (Figure 3b, S3c) should read (Figure 2d, S4c)

3- On page 11, lines 21-23: Figure 2c does not reflect the statement in the sentence

Referee #3 (Remarks to the Author):

The RAS-RAF-MAPK cascade is vital for growth factor cell signaling and regulates many biological processes. Previous works have shown that the MRAS-SHOC2-PP1 phosphatase complex plays critical roles in RAF activation, and germline gain-of-function mutations of this complex result in congenital RASopathy syndromes. The current manuscript by Hauseman et al. presents a 1.95 Å high-resolution X-ray crystal structure of the ternary MRAS-SHOC2-PP1c complex and reveals major interfaces of this important complex, which should be technically solid. The crystal structure and related biochemical analysis provides key insights into how some of the SHOC2, MRAS mutants may lead to RASopathy syndromes. This work also provide evidence that GTP-bound active form of RAS in particular M-RAS is critical for formation of the MRAS-SHOC2-PP1c complex. However, it remains unclear how PP1c dephosphorylates RAF

We thank the reviewer for pointing out the solidity of our structural findings and the key insights provided to the field for the understanding of the (to date) incurable Rasopathies syndromes.

Specific issues:

1. Based on the crystal structure, authors proposed that “two independent RAS-GTP molecules

must independently recruit RAF-14-3-3 and SHOC2-PP1c ...” (in the abstract). It may be premature to emphasize “independently” without providing direct evidence for this. In fact, a RAS-RAF signalosome model has been recently proposed in which dimerization/oligomerization of KRAS plays a key role in RAS-RAF interaction (Mysore et al. NSMB 2021). The statement in the current abstract is questionable.

We appreciate this comment and agree that the statement in the initial submission may be premature. We will remove “independently” from this sentence. To clarify, our findings clearly indicate that one individual RAS molecule cannot recruit RAF-14-3-3 and the SHOC2-PP1 holophosphatase simultaneously, hence two ‘individual’ RAS molecules are needed. We also have addressed similar concerns from the other two referees with additional experimentation demonstrating that RAF-RBD is mutually exclusive with SHOC2 for binding to activated RAS (please see **new Fig, 4b, c, d, previously shown in this rebuttal to address Referee #1, point #2**) But, as the referee has correctly pointed out, we do not provide any evidence to show co-localization vs dimerization/oligomerization, so the term “independently” is indeed inadequate.

2. The N-terminal truncated SHOC2(80-582) is used for crystallographic studies. This should be clarified in the main text and main figure legends. Was SHOC2(80-582) or FL-SHOC2 used in other biochemical (in particular binding) analysis? Please clarify.

We apologize for the lack of clarity. We have used the 80-582 construct that was successfully resolved in the X-ray ternary structure in all biochemical experiments across the manuscript, with the exception of the newly performed experiments with the FL requested by this referee (**New Figure 1a and 5a** “SHOC2 FL, residues 2-582”, **previously shown in this rebuttal to address Referee #1, point #2**) and new TR-FRET assay (**new Fig, 4b, c, d, previously shown in this rebuttal to address Referee #1, point #2**). We also added additional text to explicitly address where SHOC2 80-582 or FL was used for clarity.

3. Would the missing N-terminal region of SHOC2 play a role in forming the MRAS-SHOC2-PP1c complex?

We now add additional data that explicitly measures the contribution of the SHOC2 N-terminus (**New Figure 1a and 5a, as in point above**). We find that the SHOC2 N-terminus does contribute to ternary complex formation but not to RAS-SHOC2 binary interaction, suggesting a possible direct contact with PP1C.

4. In Table 1, at least for crystallographic data collection, highest resolution bin parameters should be shown separately.

We amended Extended Data table 1 to include this data explicitly.

5. Given the resolution and crystallographic statistics, the crystal structural model should be reliable. For Ramachandra plot outlier residues (Table 1), to evaluate if their deviations from standard stereochemistry are functionally relevant, their positions in the complex structure and related electron density maps should be provided in Supplementary Info.

We have added the list of Ramachandran outliers here:

Subunit Residue Chain

PP1C Asp95 C,F

PP1C Ser224 C,F
PP1C His248 C
SHOC2 Gly349 E
SHOC2 Asp166 B
MRAS Asp9 (Chain A)
PP1C Tyr144 (Chain C & F)
PP1C Ala247 (Chain F)
MRAS SerA (Chain A)
SHOC2 Asn389 (Chain E)
SHOC2 Asn248 (Chain E)

SHOC2 Asn179 (Chain B,E)
Shoc2 Asn202 (Chain B,E)
Shoc2 Asn133 (Chain E)
Shoc2 Asn271 (Chain B)
Shoc2 Asn225 (Chain B)

6. Table 1, Completeness listed twice.

We have amended this with the new Extended Data Table 1.

7. Sup. Fig 1., color spectrum bar indicating conservation level should be included;

We have added a bar to the extended data figure.

References:

1. Snead, K., et al., *Polycistronic baculovirus expression of SUGT1 enables high-yield production of recombinant leucine-rich repeat proteins and protein complexes*. *Protein Expr Purif*, 2022. **193**: p. 106061.
2. Sulahian, R., et al., *Synthetic Lethal Interaction of SHOC2 Depletion with MEK Inhibition in RAS-Driven Cancers*. *Cell Rep*, 2019. **29**(1): p. 118-134 e8.

3. Lai Lick, P., et al., *Classical RAS proteins are not essential for paradoxical ERK activation induced by RAF inhibitors*. Proceedings of the National Academy of Sciences, 2022. **119**(5): p. e2113491119.
4. Young, L.C., et al., *SHOC2–MRAS–PP1 complex positively regulates RAF activity and contributes to Noonan syndrome pathogenesis*. Proceedings of the National Academy of Sciences, 2018. **115**(45): p. E10576.

Reviewer Reports on the First Revision:

Referees' comments:

Referee #1 (Remarks to the Author):

The authors successfully addressed most of my concerns and the study is now acceptable for publication.

Referee #2 (Remarks to the Author):

The authors responded satisfactorily to all my comments.

Referee #3 (Remarks to the Author):

Authors have addressed most of my major concerns. However, with high resolution of the reported crystal structure, authors still observed many Ramachandra plot outlier residues. Surprisingly, instead of explaining if deviations of these residues from standard stereochemistry are functionally relevant, authors simply removed the required Ramachandra plot parameters from the Table 1. Other than this, additional data provided in this revised revision clarified some of important issues, although it remains unclear how the N-terminal region of SHOC2 contributes to MRAS-SHOC2-PP1c complex formation and signaling.